# Effects of acidification on nitrification and associated nitrous oxide emission in estuarine and coastal waters

Jie Zhou [1,7], Yanling Zheng [1,2,3,4,7] ✉, Lijun Hou [1] ✉, Zhirui An [1], Feiyang Chen [1], Bolin Liu [1], Li Wu [2,3], Lin Qi [2,3], Hongpo Dong [1], Ping Han [2,3,4], Guoyu Yin [2,3,4], Xia Liang[1], Yi Yang [2,3,4], Xiaofei Li [1], Dengzhou Gao [1], Ye Li [2,3,4], Zhanfei Liu[5], Richard Bellerby[6] & Min Liu [2,3,4] ✉

In the context of an increasing atmospheric carbon dioxide ($CO_2$) level, acidification of estuarine and coastal waters is greatly exacerbated by land-derived nutrient inputs, coastal upwelling, and complex biogeochemical processes. A deeper understanding of how nitrifiers respond to intensifying acidification is thus crucial to predict the response of estuarine and coastal ecosystems and their contribution to global climate change. Here, we show that acidification can significantly decrease nitrification rate but stimulate generation of byproduct nitrous oxide ($N_2O$) in estuarine and coastal waters. By varying $CO_2$ concentration and pH independently, an expected beneficial effect of elevated $CO_2$ on activity of nitrifiers ("$CO_2$-fertilization" effect) is excluded under acidification. Metatranscriptome data further demonstrate that nitrifiers could significantly up-regulate gene expressions associated with intracellular pH homeostasis to cope with acidification stress. This study highlights the molecular underpinnings of acidification effects on nitrification and associated greenhouse gas $N_2O$ emission, and helps predict the response and evolution of estuarine and coastal ecosystems under climate change and human activities.

Carbon dioxide ($CO_2$) in the atmosphere has been increasing due to intensive human activities such as combustion of fossil fuels, cement production, deforestation, and other land-use changes[1]. Globally, the average atmospheric concentration of $CO_2$ has now reached 413.2 ppm and is expected to exceed 800 ppm by the end of the 21st century[2,3]. Approximately 40% of the emitted $CO_2$ during the industrial era has been absorbed by the oceans[4], consequently causing a reduction of about 0.1 pH unit in surface seawater[5,6]. A further decline of 0.2–0.3 pH units is estimated at the end of this century, with severe consequences expected for sensitive organisms and ecosystems[6–8].

Estuarine and coastal ecosystems are dynamic regions under the interaction of rivers, land, and oceans[9], which can provide vital ecosystem services for human well-being[10]. In the context of an increasing atmospheric $CO_2$ level, estuarine and coastal waters, however, suffer from more acute acidification than open oceans, under the synergistic effects of land-derived nutrient inputs, coastal upwelling, and complex biogeochemical processes (Supplementary Fig. 1)[11,12]. One of the greatest threats to estuarine and coastal ecosystems worldwide is the excess input of watershed anthropogenic nutrients[10]. The eutrophication-induced phytoplankton production can result in high

[1]State Key Laboratory of Estuarine and Coastal Research, Yangtze Delta Estuarine Wetland Ecosystem Observation and Research Station, East China Normal University, Shanghai 200241, China. [2]School of Geographic Sciences, East China Normal University, Shanghai 200241, China. [3]Key Laboratory of Geographic Information Science (Ministry of Education), East China Normal University, Shanghai 200241, China. [4]Key Laboratory of Spatial-temporal Big Data Analysis and Application of Natural Resources in Megacities, Ministry of Natural Resources, Shanghai 200241, China. [5]The University of Texas at Austin Marine Science Institute, Port Aransas, TX 78373, USA. [6]Norwegian Institute for Water Research, Thormøhlensgt 53D, 5006 Bergen, Norway. [7]These authors contributed equally: Jie Zhou, Yanling Zheng. ✉e-mail: ylzheng@geo.ecnu.edu.cn; ljhou@sklec.ecnu.edu.cn; mliu@geo.ecnu.edu.cn

respiration rate in bottom waters where the algal-derived matter settles, which may cause strong $CO_2$ production[13]. Acidification in estuarine and coastal waters can thus be greatly intensified by episodic intrusion of high-$CO_2$ upwelled water[11,13,14], which may detrimentally affect biological processes and functioning of estuarine and coastal ecosystems[15–21].

Nitrification is a critical process for the balance of reduced and oxidized nitrogen pools, linking mineralization to nitrogen removal pathways of denitrification and anaerobic ammonium oxidation[22]. It thus plays a crucial role in the global nitrogen cycle, especially in eutrophic aquatic ecosystems. Due to the slow growth of nitrifiers and their high sensitivity to environmental perturbations[23], nitrification is anticipated to be disturbed by aquatic acidification. One complication in the response of nitrifiers to acidification is that the increase of partial pressure of $CO_2$ ($pCO_2$) and the decrease of pH may have opposing effects. Higher $pCO_2$ condition is expected to benefit nitrification, as an increased carbon source may promote the growth of chemoautotrophic nitrifiers ($CO_2$-fertilization)[24–27]. In contrast, the concomitant decrease in pH can shift the equilibrium between ammonia ($NH_3$) and ammonium ($NH_4^+$) toward a lower concentration of substrate $NH_3$ available for ammonia-oxidizers and thereby inhibit nitrification[25,28,29]. The response of nitrifiers thus depends strongly on the balance of these potential positive and negative effects. However, little is known concerning the effects of projected levels of aquatic acidification on the metabolisms of nitrifiers and underlying mechanisms.

Nitrification is also an important pathway for production of greenhouse gas nitrous oxide ($N_2O$)[30–33], which has >300-fold stronger radiative forcing per mole than $CO_2$ and can react with ozone in the stratosphere[34]. $N_2O$ can be enzymatically produced by ammonia-oxidizing bacteria (AOB) via conversion of hydroxylamine ($NH_2OH$) to $N_2O$[35,36], or via nitrifier denitrification [a sequential reduction of nitrite ($NO_2^-$) to nitric oxide (NO) and $N_2O$][35]. Recently, the biotic conversion of $NH_2OH$ to $N_2O$ by AOB through the cytochrome P460 was also characterized[36]. In contrast, $N_2O$ emission associated with $NH_3$ oxidation by ammonia-oxidizing archaea (AOA) is believed to result mainly from abiotic reactions between $NH_2OH$ and $NO_2^-$ or NO[32,37,38]. Previous studies suggested that AOA produce lower yields of $N_2O$ than AOB during aerobic $NH_3$ oxidation[37,39]. In addition, it has been documented that complete ammonia oxidizers (comammox) exhibit a lower $N_2O$ yield than AOB, as $N_2O$ originates rather from the abiotic conversion of $NH_2OH$ by comammox bacteria[40,41]. However, it is not yet clear how nitrifier $N_2O$ production will respond to aquatic acidification.

Here we examine how aquatic acidification affects nitrification rate and associated $N_2O$ emission in the Yangtze Estuary and adjacent coastal waters. Manipulation experiments are also conducted to decouple the individual effects of elevated $pCO_2$ and reduced pH. Metatranscriptomes are further analyzed to elucidate the metabolic response of nitrifying microbes by tracking the expression of acidification responsive genes. This research provides insights into the mechanistic interactions between acidification and nitrification, and helps predict the future ecological evolution of estuarine and coastal ecosystems.

## Results and discussion
### Effects of acidification on nitrification rate
Different acidification levels (pH reduced by 0.10–1.05) were achieved via bubbling water samples collected from six representative sites (Yz1–Yz6) along the Yangtze Estuary and adjacent coastal waters with different air:$CO_2$ gas mixtures (Fig. 1, Supplementary Tables 1 and 2). Nitrification rates at these sites showed an identical response to acidification: all decreased remarkably when pH was reduced regardless of a potential beneficial effect of high $CO_2$ (Supplementary Fig. 2). A decrease in nitrification rates (5.8–18.1%) was detected under pH reduction even from 7.92–8.15 to 7.80–8.04 ($pCO_2$ increased by 122–172 µatm) ($P < 0.05$). Nitrification rates decreased by -11.1–34.1% when the $pCO_2$ was doubled ($P < 0.05$). The decrease of nitrification rates was strongly correlated with the reduction of water pH ($P < 0.05$), based on the constructed acidification–response curves (Fig. 2a). Nitrification rates would decrease by -7.7–25.0% under an average reduction of about 0.21 pH units which has been observed in estuarine and coastal waters across the world over the past several decades (Supplementary Fig. 1). This inhibition effect of acidification on nitrification rate is consistent with what was previously observed in the open oceans[28,42]. Whilst nitrification rate was reported to increase along a decreasing natural gradient of pH in Narragansett Bay[43], it was likely due to a combination of biogeochemical conditions rather than the effect of acidification alone.

The inhibition of nitrification rate by acidification tended to be lower in the upper estuary waters where AOB were the dominant ammonia oxidizers than that in the adjacent coastal regions where the ammonia-oxidizing communities were dominated by AOA (Fig. 2a and Supplementary Fig. 3). In addition to the heterogeneity of nitrifying microbial communities, this variability in the influence of acidification on nitrification rate may stem from multiple biogeochemical factors[43]. Especially, the relatively higher $NH_4^+$

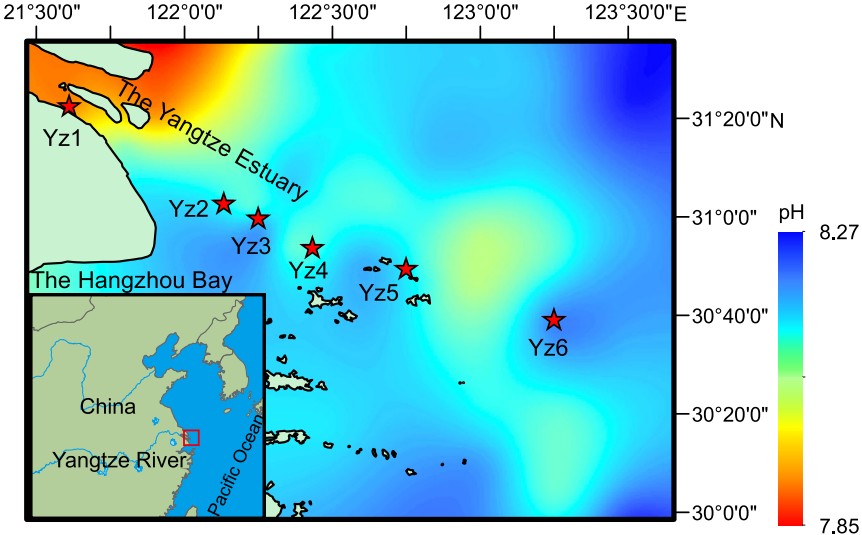

**Fig. 1 | Study area and sampling locations overlaid on pH values of near-bottom water.** Stations are marked by red stars.

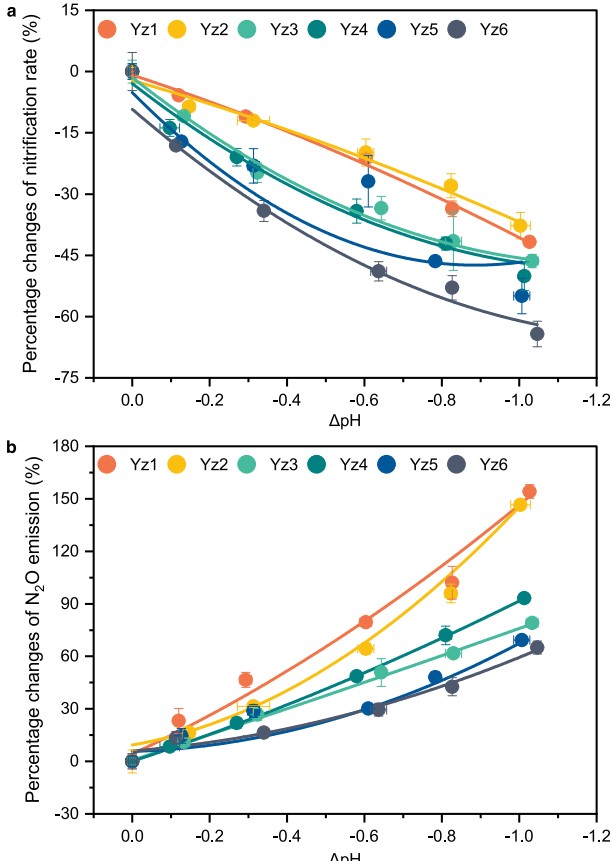

**Fig. 2 | Response of nitrification and associated N$_2$O production rates to simulated aquatic acidification in Yangtze Estuary and its adjacent coastal area. a** Nitrification rates. Data show the percentage changes of nitrification rates in the acidified treatments compared to the ambient control. For all the lines, $P < 0.05$. **b** N$_2$O production rates. Data show the percentage changes of N$_2$O production rates in the acidified treatments compared to the ambient control. For all the lines, $P < 0.05$. Error bars represent SD ($n = 3$ biologically independent samples). $\Delta$pH corresponds to the decrease between water pH before and after acidification. The fitting curve was obtained by polynomial fitting method. Equations and $P$ values for the fitted curves are given in Supplementary Table 3. Source data are provided as a source data file.

concentrations in the upper estuaries may mitigate the inhibiting effects on ammonia oxidation caused by acidification (Supplementary Table 1). Consistently, the measured inhibition of nitrification rate by acidification in the estuarine and coastal waters was generally lower than that in the oligotrophic seas where nitrification rates were reported to decline by 3–44% in response to a decrease of 0.1 pH unit[28,42]. Indeed, NH$_4^+$ concentration was negatively correlated with the inhibition effect of acidification on nitrification rate in different habitats ranging from estuary to open ocean ($P < 0.05$)[28,42,44] (Supplementary Fig. 4). Nevertheless, considering the faster and continuous aggravation of acidification in estuarine and coastal waters due to the synergistic effects of human activity-induced eutrophication and elevated atmospheric CO$_2$ level[11], its disturbance on nitrification rates could cause profound consequences on estuarine and coastal ecosystem processes.

### Effects of acidification on associated N$_2$O production
Based on the response of nitrification rates, ref. [28] speculated that the decrease in nitrification rates as a result of ocean acidification could greatly reduce N$_2$O production in the open ocean. The impact of aquatic acidification on the N$_2$O production, however, may be decoupled from its impact on the nitrification rates[45]. For

example, ref. [42] recently reported that when seawater pH in the western North Pacific was reduced, the N$_2$O production increased significantly while nitrification rates remained stable or even decreased. However, in their work, the acidification was manipulated by adding strong acid (HCl). Although the addition of HCl can elevate $p$CO$_2$ and reduce pH, it also alters alkalinity and results in different carbonate parameters compared with those expected in the future, i.e., dissolved inorganic carbon (DIC) increases under natural aquatic acidification rather than remains unchanged[46]. Moreover, nitrifying communities in diverse aquatic habitats may respond differently to acidification, as the mechanisms for N$_2$O production differ among different nitrifying organisms[32,38,40]. Therefore, the response of N$_2$O production during nitrification to aquatic acidification in estuarine and coastal waters needs to be evaluated.

Through aerating with sterile air at different CO$_2$ levels[47], which can best mimic the ongoing aquatic acidification, we found that the production rates of N$_2$O at all sampling sites were enhanced significantly by acidification (Supplementary Fig. 5). Even under a pH decline of -0.1 unit (7.92–8.15 to 7.80–8.04), a significant promotion of N$_2$O production (8.4–23.1%) was observed at the end of incubation ($P < 0.05$). Acidification–response curves were constructed between the decline in pH and N$_2$O emission, showing a significant increase of N$_2$O production rates along with the increase of acidification levels ($P < 0.05$) (Fig. 2b). Under an average reduction of about 0.21 pH units detected in estuarine and coastal regions worldwide, the rates of N$_2$O production during nitrification might increase by ~9.5–27.5% (Fig. 2b). These findings support our hypothesis that, similar to other environmental perturbations such as low oxygen and toxicant exposures[23,42,48], acidification can increase N$_2$O production in estuarine and coastal waters, regardless of whether AOB or AOA dominated. Therefore, although the mechanisms for N$_2$O production differ among different nitrifying organisms[32,38,40], their N$_2$O production rate might increase under pH reduction. The increased N$_2$O production under acidified conditions in estuarine and coastal waters is consistent with those in the western North Pacific[42]. However, ref. [44] documented inhibition of N$_2$O production by ocean acidification in cold temperate and polar seawaters. Assuming that nitrification is the main N$_2$O production pathway in their study, the response of the N$_2$O production to acidification would be different in polar seas. Although heterotrophic denitrifiers may also contribute to the production of N$_2$O, their contribution may be insignificant, as the natural isotopic signatures of N$_2$O showed that the pathway of NO$_2^-$ reduction (including nitrifier denitrification and heterotrophic denitrification) contributed only 0–13.3% of the released N$_2$O (Supplementary Table 4). Moreover, the samples from all study sites were well oxygenated [dissolved oxygen (DO): 8.30–9.86 mg L$^{-1}$; Supplementary Table 1] and remained at high DO levels during the incubation (Supplementary Table 2), which was unlikely to occur for heterotrophic denitrification. Previous studies reported that when the DO concentration is more than 0.06 mg L$^{-1}$, the N$_2$O production by heterotrophic denitrification is completely inhibited[49]. Therefore, the contribution of denitrifying bacteria to the production of N$_2$O should be negligible. This study demonstrates that N$_2$O production during nitrification of both AOB-dominated and AOA-dominated nitrifiers can be stimulated by acidification, which is an important step for evaluating the impact of ongoing acidification on N$_2$O emission in the complex and dynamic estuarine and coastal ecosystems. Should our results be representative and that nitrifiers contribute to half of global estuarine and coastal N$_2$O emissions[50,51] (Supplementary Fig. 6 and Supplementary Table 5), nitrification-derived N$_2$O emission in these ecosystems would increase by 0.05–0.15 Tg N$_2$O-N yr$^{-1}$ in response to an average decrease of 0.21 pH units (Supplementary Text 1), accounting for 0.7–2.2% of the total anthropogenic N$_2$O emissions globally (6.9 Tg N$_2$O-N yr$^{-1}$)[52].

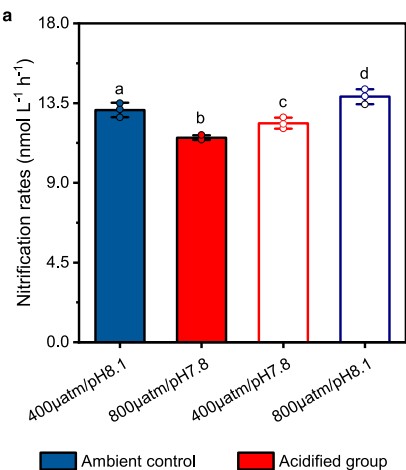
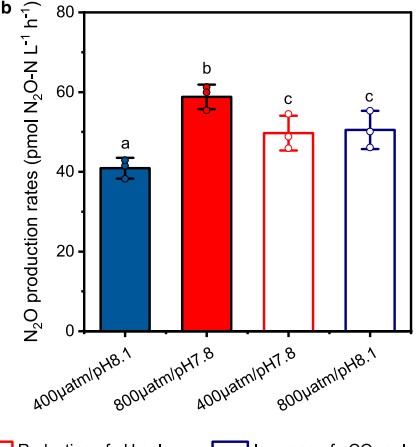

**Fig. 3 | Individual effects of increasing $p$CO$_2$ and decreasing pH on the activity of nitrifiers in estuarine and coastal waters. a** Nitrification rates. **b** N$_2$O production rates during nitrification. Four scenarios were constructed: the ambient control (400 μatm/pH 8.1, blue solid bars), the acidified group (800 μatm/pH 7.8, red solid bars), reduction of pH only (400 μatm/pH 7.8, red open bars), and increase of $p$CO$_2$ only (800 μatm/pH 8.1, blue open bars). Different lowercase letters (a, b, c and d) above the columns indicate significant differences among the four scenarios ($P < 0.05$). Error bars represent SD ($n = 3$ biologically independent samples), and dots are corresponding data points of the replicates. Significant differences were determined via one-way analysis of variance (ANOVA). Source data are provided as a source data file.

## Decoupling potential effects of elevated $p$CO$_2$ and reduced pH

Our results showed that acidification can inhibit nitrification rates but enhance N$_2$O emissions in estuarine and coastal waters (Fig. 2). However, the increase in $p$CO$_2$ and decrease in pH may have opposing effects on nitrifiers. To distinguish the individual effects of elevated $p$CO$_2$ and reduced pH, a series of open, continuous-flow microcosm systems were constructed. The carbonate chemistry was manipulated by steadily bubbling collected water samples from site Yz3 with CO$_2$ adjusted air (400 μatm and 800 μatm) while adjusting pH (7.8 and 8.1) with sterile acid or base solution according to a real-time pH detector (Supplementary Fig. 7). Equilibrium states were achieved in four scenarios [low $p$CO$_2$ (400 μatm) and high pH (8.1); low $p$CO$_2$ (400 μatm) and low pH (7.8); high $p$CO$_2$ (800 μatm) and low pH (7.8); high $p$CO$_2$ (800 μatm) and high pH (8.1)] and remained stable during the incubation period (Supplementary Table 6). Results showed that nitrification rates significantly decreased at low pH, independent of $p$CO$_2$ level (Fig. 3a), demonstrating the negative effects of reduced pH on nitrification rates. However, the effect of elevated $p$CO$_2$, which is expected to benefit nitrification, seems to be pH-dependent (Fig. 3a). When pH was maintained at the ambient 8.1, an obvious "CO$_2$-fertilization" effect was observed as nitrification rates increased at high $p$CO$_2$ (Fig. 3a). In contrast, under acidified conditions (pH 7.8), elevated $p$CO$_2$ caused an unexpected decrease in nitrification rates (Fig. 3a). This pattern suggests that, when the metabolic processes of the related nitrifiers are affected by reduced pH, increased $p$CO$_2$ becomes an additive stressor. These results are in contrast to the observation from N$_2$-fixing cyanobacteria, which can benefit from high $p$CO$_2$ under reduced pH conditions[53].

N$_2$O production during nitrification was promoted under acidified conditions (Fig. 2b), by both the reduced pH and the elevated $p$CO$_2$ (Fig. 3b). The promotion of N$_2$O production under high $p$CO$_2$ while maintaining the ambient pH was unexpected (Fig. 3b) as this condition was beneficial for nitrifiers (CO$_2$-fertilization; Fig. 3a). It is possible that the production of by-product N$_2$O could increase along with the increase of nitrification rates. However, this possibility cannot fully explain the enhanced N$_2$O emission under the condition of sole $p$CO$_2$ elevation ($p$CO$_2$ 800 μatm/pH 8.1), as the promotion of N$_2$O production rate was significantly higher than the nitrification rate (Fig. 3). Under natural acidified conditions (an elevation of $p$CO$_2$ concomitant with pH reduction), the strongest promotion of N$_2$O emission may be expected, when the effects of pH reduction and $p$CO$_2$ elevation are combined (Fig. 3b). This is the first attempt to decouple the individual effects of elevated $p$CO$_2$ and reduced pH on nitrification rate and associated N$_2$O emission in acidified aquatic environments, providing insights into the underlying mechanism of aquatic acidification affecting nitrifiers. Their individual effects were successfully distinguished based on our constructed continuous-airflow and automatic pH incubation systems. It was previously speculated that increasing levels of $p$CO$_2$ may cause positive effect on the activity of chemoautotrophic nitrifiers[54]. In contrast, under acidified conditions, elevated $p$CO$_2$ may further inhibit nitrification rate and promote the undesirable by-product N$_2$O emission. Therefore, the negative effects of aquatic acidification on microbial nitrogen transformations and their feedback to global climate change are probably more intensive than previously thought (Supplementary Fig. 8).

## Transcriptional response of nitrifiers to acidification during long-term incubation

As an important molecular response to acidification stress, nitrifiers may adjust gene expressions by intracellular signaling networks[55], which can be further reflected by reduced nitrification rate and enhanced N$_2$O production. However, the transcriptional response of nitrifying communities to aquatic acidification remains unknown. Metatranscriptome sequencing can be used to interrogate the differential expression of genes involved in the physiological metabolism of nitrifying communities under acidified conditions[23]. However, previous attempts to acquire metatranscriptomes based on the short-term acidification experiments failed, because there was not enough mRNA with good integrity and purity, which might be due to the extreme instability of mRNA and the complexity of environmental samples[23]. More importantly, nitrifiers account for only a small fraction of the complex microbial communities in estuarine and coastal waters (<5%; Supplementary Fig. 9), thus it is difficult to extract sufficient nitrifying transcripts to fully reveal the physiological metabolism of nitrifiers. Therefore, a series of continuous-flow environmental simulators with water samples from site Yz3 were set up to mimic long-term acidification and to enrich nitrifiers. The pH and $p$CO$_2$ in the ambient controls were maintained at about 8.1 and 400 μatm, respectively, whereas they were stabilized at about 7.8 and 800 μatm, respectively, in the acidified treatments. After ~25 days, the continuously operated simulators exhibited stable nitrification reactions (Supplementary Fig. 10) and dominance of nitrifying communities (accounting for

44.6% and 45.5% in the ambient controls and acidified treatments, respectively) (Supplementary Figs. 11, 12). The in-situ nitrifying bacteria were simultaneously enriched, and the original members in the environment (*Nitrosomonas* and *Nitrospira*) remained dominant. However, the cell numbers of AOA were not greatly increased in the nitrifying enrichments, and thus AOA cultivation was further conducted with streptomycin to inhibit nitrifying bacteria. After 50 days of incubation, the relative abundance of AOA increased from 1.1% to 11.2%, whereas nitrifying bacteria were undetectable (Supplementary Fig. 13). For these nitrifying enrichment cultures, significant reduced nitrification rates and stimulated $N_2O$ emissions were observed in the acidified treatments (Supplementary Figs. 14, 15), a pattern consistent with that of the field water samples. Although the enrichment manipulation changes the original environmental microbial communities, the nitrifying enrichments are representatives to investigate the transcriptional response of nitrifiers to acidification in complex estuarine and coastal waters.

According to the metatranscriptomic analyses (Supplementary Table 7), $CO_2$-induced acidification can significantly affect the physiological metabolisms of nitrifiers at transcriptional level, as those genes involved in nitrogen transformations, cytosolic pH homeostasis, energy generation, and $CO_2$ fixation were all greatly regulated by acidification (Figs. 4a–c, 5a–c). Expression of the potentially active subunit of the ammonia monooxygenase gene (*amoA*) of AOB was down-regulated by 66% for the acidified treatments (Fig. 4a). Real-time quantitative polymerase chain reaction (qPCR) also demonstrated that the expression of bacterial *amoA* gene was significantly down-regulated under acidified conditions ($P < 0.01$; Supplementary Table 9). Consistently, expressions of bacterial *amoB* (ammonia monooxygenase subunit B, also suggested as a catalytic subunit[56]) and *amoC* (ammonia monooxygenase subunit C) decreased by 68 and 69%, respectively, after acidification (Fig. 4c). In addition, expression of hydroxylamine dehydrogenase gene (*hao*) was down-regulated by 61% ($P < 0.01$), while the expressions of nitrite oxidoreductase genes *nxrA* and *nxrB* were reduced by 76 and 61% ($P < 0.01$), respectively, under acidified conditions (Fig. 4c and Supplementary Table 9). Based on the transcriptional data of AOA, expression of archaeal *amoA* was also down-regulated (24%) under acidified conditions ($P < 0.05$) (Fig. 5a and Supplementary Table 9). Additionally, transcripts of archaeal *amoC* were down-regulated by 43%, while the expression of archaeal *amoB* remained unchanged under acidification (Fig. 5c). Overall, the functional gene transcripts involved in the stepwise oxidation of $NH_3$ to nitrate ($NO_3^-$) were generally down-regulated under acidified conditions, consistent with the reduction of nitrification rates (Fig. 2a).

In contrast, transcripts of genes encoding nitrifying bacterial nitric oxide reductase ($N_2O$-forming, nor) were up-regulated by acidification (Fig. 4a). The expressions of *norB*, *norC*, *norD*, and *norQ* genes of AOB were up-regulated by 2.5, 16.2, 0.7, and 0.2 folds, respectively (Fig. 4c and Supplementary Table 9). These results provide molecular evidence for the observed stimulation of $N_2O$ emission under acidified conditions. Although alternative enzymes might have also contributed to nitrifying bacterial $N_2O$ emissions, such as cytochrome c554 (encoded by *cycA*)[57], cytochrome c′-β (encoded by *cytS*)[58], and cytochrome P460[36], transcripts of these proteins were not observed in this study. In contrast to the dramatically up-regulated expression of nitric oxide reductase genes, transcripts of nitrite reductase (NO-forming, nirK) of nitrifying bacteria were down-regulated by 43% in the acidified treatments compared with the ambient control ($P < 0.05$) (Fig. 4c and Supplementary Table 9), implying that $NO_2^-$ reduction by nitrifying bacteria might have been inhibited by acidification. Therefore, the enhanced emission of $N_2O$ under acidified conditions was not sourced from the denitrification pathway of bacterial nitrifiers. Although it was suggested that the abiotic hybrid reaction was the main source for archaeal $N_2O$ yield[32,37,38], presumptive enzymes including copper hydroxylamine oxidoreductase (Cu-HAO) and putative nitroxyl

oxidoreductase were speculated to be involved in the $N_2O$ production of AOA[59]. However, transcripts of these presumptive proteins were not observed (Fig. 5a). In addition, archaeal copper nitrite reductase (nirK) may function as bacteria-like HAO to oxidize $NH_2OH$ to $N_2O$, or may be involved in the $N_2O$ production of AOA via nitrifier denitrification[59]. Nevertheless, the transcriptional response of archaeal *nirK* gene may not be the main cause of the increased emission of $N_2O$ under acidification, as the expression of archaeal *nirK* was slightly down-regulated (Fig. 5c). On the contrary, transcripts of the presumptive archaeal nitric oxide reductase *norQ* gene were up-regulated by 0.2 folds under acidified conditions (Fig. 5c). Whereas, due to the limited understanding of AOA-driven $N_2O$ production pathways, the molecular mechanism of AOA-mediated elevation of $N_2O$ emission under acidification needs to be further elucidated.

The intracellular pH of cyanobacterium was reported to decrease along with water pH under acidified conditions[53]. If the case happens for nitrifiers, they may need to generate more proton motive force (PMF), which is required for adenosine triphosphate (ATP) synthesis[60]. However, the gene transcripts involved in the proton-translocating membrane-bound enzymes of the enriched nitrifying bacteria were down-regulated in the acidified treatments ($Log_2FC < -1$) (Fig. 4a). Nitrifying bacterial transcripts of NADH-quinone oxidoreductase gene (*nuo*) of complex I, ubiquinol-cytochrome c reductase gene (*pet*) of complex III, and cytochrome c oxidase gene of complex IV were down-regulated by 60%, 57%, and 61%, respectively, under acidified conditions (Fig. 4c). Thus, based on these data, we cannot distinguish whether the cytoplasmic pH of nitrifying bacteria was reduced under the imposed degree of acidification. Nevertheless, an important insight was obtained when probing into the transcripts of genes encoding the energy-yielding adenosine triphosphatases (ATPases). Bacterial ATPase family comprises membrane-bound protein complexes responsible for either ATP synthesis using a cross-membrane PMF, or establishing PMF using the energy released from ATP hydrolysis[61,62]. Based on the metatranscriptomic data, transcripts of genes encoding the bacterial V-type ATPases, working as ATP-dependent proton pumps, were up-regulated (up to 11.5-fold increase) under acidified condition (Fig. 4c). This result suggests an increasing need of nitrifying bacteria for pumping cytoplasmic protons to maintain pH homeostasis and the $H^+$ gradient under a reduction of -0.3 pH units. Similarly, transcripts of archaeal *nuo* gene of complex I, cytochrome c oxidase *cox* gene of complex IV, and archaeal V-type ATPase *atp* genes were significantly up-regulated ($Log_2FC > 0.5$; Fig. 5a), showing that the intracellular pH of the AOA may have also decreased with water pH.

Under acidified conditions, an increasing energy demand for translocation of substrates across membranes might occur, because gene transcripts encoding the corresponding ATP-binding cassette transporters were up-regulated (with a maximum of 11-fold upregulation, Figs. 4c and 5c). These results suggest that more of the energy derived from $NH_3$ or $NO_2^-$ oxidation may have been allocated to cope with acidification stress, such as to maintain cytosolic pH homeostasis and substrate transports. However, the total production of ATP was probably reduced, as nitrification rates were significantly inhibited by acidification. In addition, transcripts of genes encoding bacterial F-type ATPases, which function as PMF-driven ATP synthases, were down-regulated by acidification (with an average reduction of 63%, Fig. 4c). Nevertheless, the expression of genes associated with the carbon-concentrating mechanism that saturates the carboxylating enzyme, ribulose bisphosphate carboxylase oxidase (Rubisco), was down-regulated under acidified conditions ($Log_2FC < -1$; Fig. 4a), suggesting a reduced energetic requirement for $CO_2$ enrichment[63] (Supplementary Text 2). This regulating mechanism may explain the observed "$CO_2$-fertilization" effect under high $p$CO_2$ but under ambient pH level (800 μatm/pH 8.1) (Fig. 3a), as more energy was probably saved and reallocated to other cellular processes. However, the saved

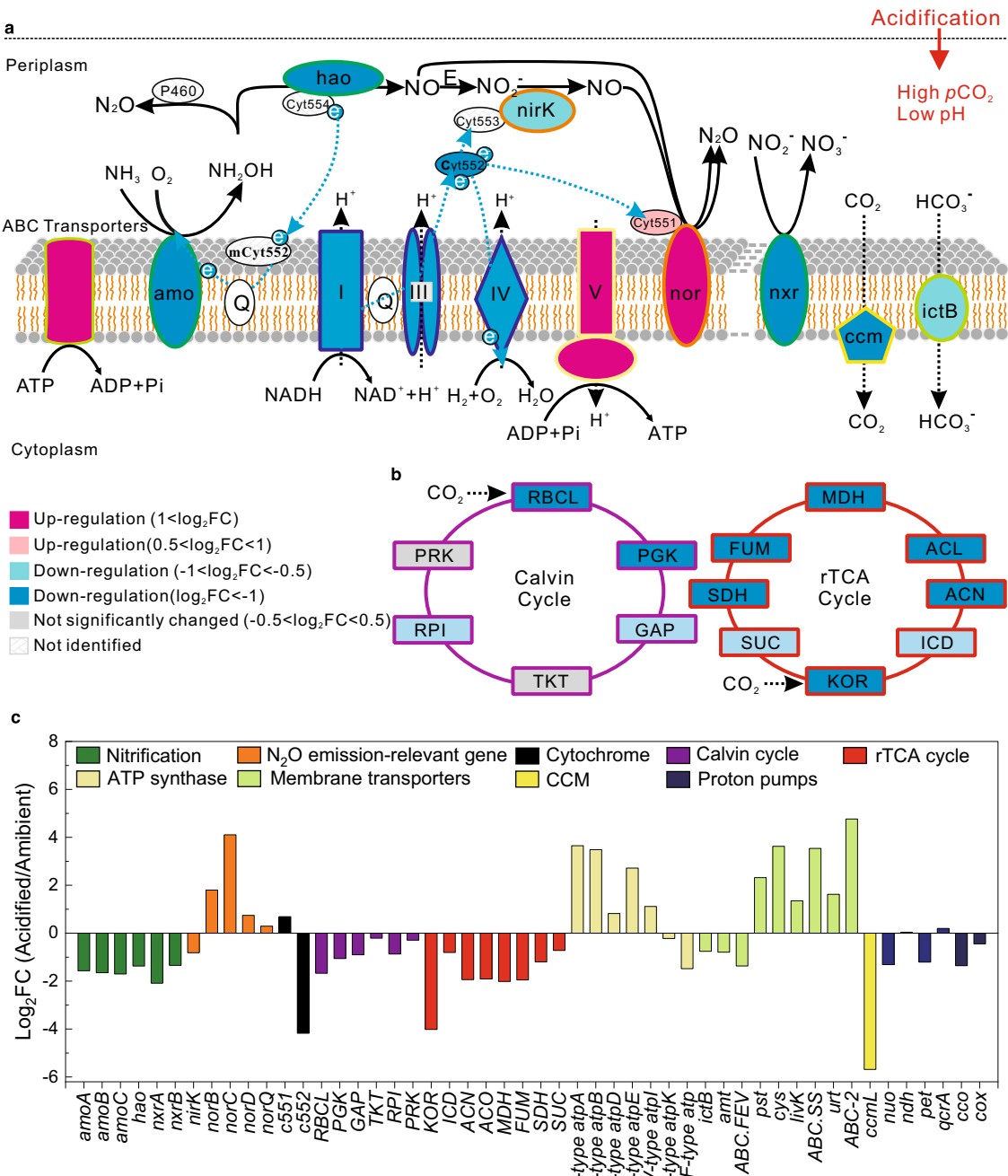

**Fig. 4 | Response of bacterial nitrifiers to acidification. a** Schematic model depicting the effects of acidification on the expression of genes involved in the stepwise oxidation of $NH_3$ ($NH_3 \rightarrow NH_2OH \rightarrow NO \rightarrow NO_2^- \rightarrow NO_3^-$), $N_2O$ production, energy generation, and cytosolic pH homeostasis of nitrifying bacteria. The membrane is broken by dotted line, as nitrite oxidation does not often occur in the same organism with ammonia oxidation, with the exception of recently discovered comammox *Nitrospira*. Fold change (FC) in relative gene expression was calculated by comparing the acidified samples to the ambient control. The roman numbers refer to the enzyme complex I (NADH dehydrogenase), complex III (cytochrome bc1 complex), complex IV (cytochrome c oxidase), and complex V (ATP synthase) in the respiratory chain. Dotted blue arrows show the movement of electrons. E, unknown enzyme. **b** Effects of acidification on the expression of genes involved in $CO_2$ fixation through Calvin and reductive tricarboxylic acid (rTCA) cycles. Colors at the center of the protein pictograms indicate the extent of up- or down-regulation of gene transcripts encoding these proteins. Colors at the periphery of the protein pictograms correspond to bar colors in (**c**). **c** FC of transcripts encoding proteins showed in (**a**) and (**b**). Definitions of the abbreviations are shown in Supplementary Table 8. Source data are provided as a source data file.

energy due to elevated carbon source seemed to be minor when compared with the disturbances caused by acidification, as significantly negative effects were observed under acidified conditions (800 μatm/pH 7.8) (Fig. 3a). Collectively, these results suggest that acidification may lead to lower production of ATP and reallocation of this energy to support cell maintenance rather than to fuel chemoautotrophic growth. Indeed, the growth rate of the dominant nitrifying bacteria was reduced at acidified treatments (reduction of ~25% and 27% for *Nitrosomonas* and *Nitrospira*, respectively) (Supplementary Fig. 16a,b). Further evidence was observed at the transcriptional level, as gene transcripts involved in $CO_2$ fixation [Calvin cycle for *Nitrosomonas* and reductive tricarboxylic acid cycle for *Nitrospira*] were ubiquitously down-regulated (Supplementary Text 3, Fig. 4b,c)[25,27]. However, gene transcripts involved in 3-hydroxypropionate/4-hydroxybutyrate carbon-fixation pathway of AOA, which is more energetically favorable[24,26], were generally up-regulated under acidified

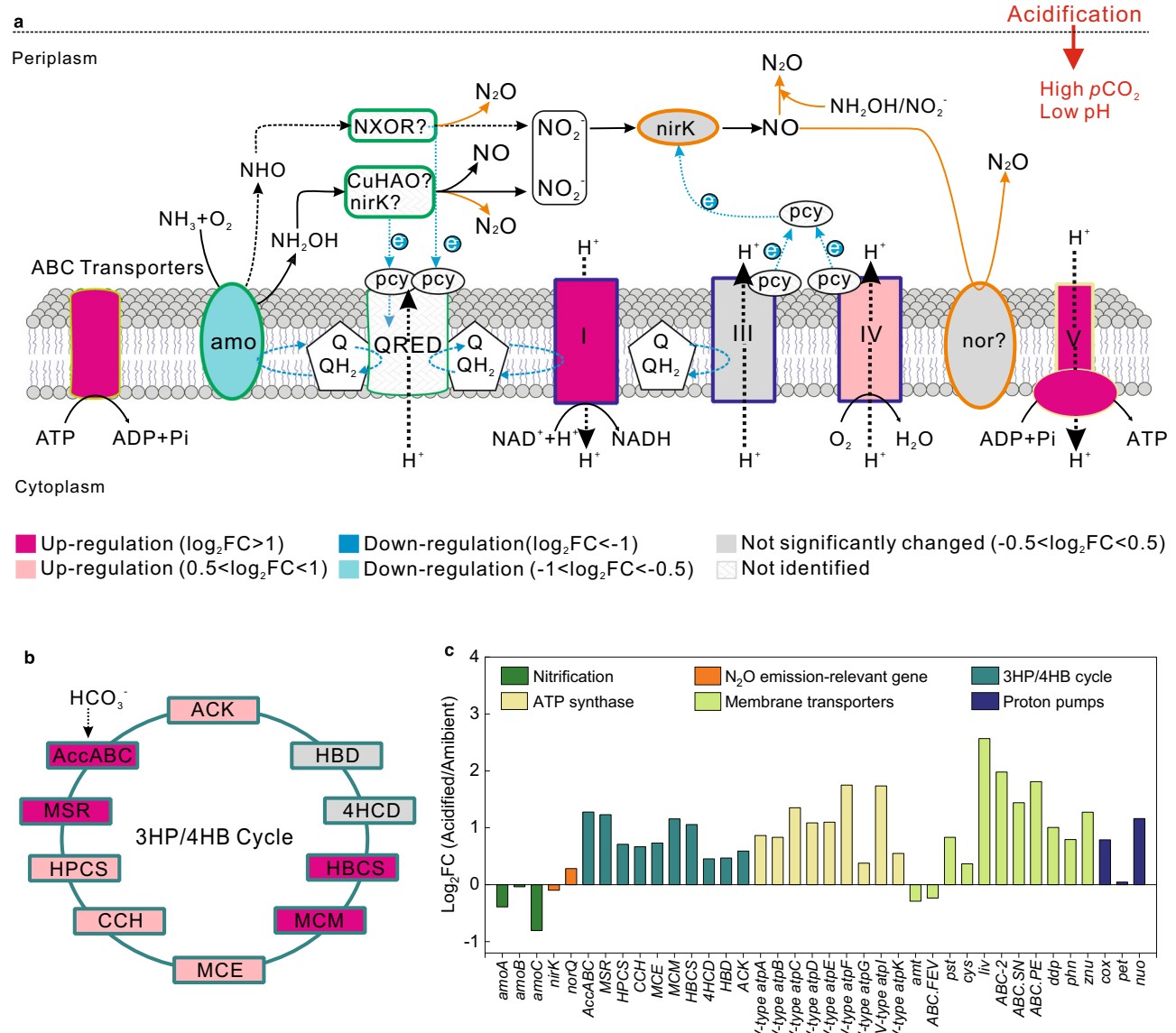

**Fig. 5 | Response of ammonia-oxidizing archaea (AOA) to acidification.**
**a** Schematic model depicting the effects of acidification on the expression of genes involved in $NH_3$ oxidation, $N_2O$ production, energy generation, and cytosolic pH homeostasis of AOA. Fold change (FC) in relative gene expression was calculated by comparing the acidified samples to the ambient control. The roman numbers refer to the enzyme complex I (NADH dehydrogenase), complex III (*b*-pcy), complex IV (pcy-aa₃), and complex V (ATP synthase) in the respiratory chain. Dotted blue arrows show the movement of electrons. **b** Effects of acidification on the expression of genes involved in 3-hydroxypropionate/4-hydroxybutyrate cycle. Colors at the center of the protein pictograms indicate the extent of up- or down-regulation of gene transcripts encoding these proteins. Colors at the periphery of the protein pictograms correspond to bar colors in (**c**). **c** FC of transcripts encoding proteins showed in (**a**) and (**b**). Definitions of the abbreviations are shown in Supplementary Table 8. Source data are provided as a source data file.

conditions (Fig. 5b,c). Consistent with these results, relatively higher AOA abundance was detected in the acidified treatments ($P < 0.05$) (Supplementary Fig. 16c), due mainly to the chemoautotrophic growth of *Nitrosopumilus* (the dominant AOA genus) whose relative abundance also increased under acidified conditions ($P < 0.05$) (Fig. 6a). Thus, aquatic acidification might change the community of ammonia oxidizers towards an increasing ratio of AOA over AOB. This variation might also be caused by higher $NH_3$ affinity of AOA[22,64] and the reduced $NH_3$ availability under acidified conditions[25]. However, ref. [44] reported that AOA assemblage composition was not sensitive to ocean acidification, possibly because the incubation period (less than one week) in their study was not long enough to cause significant turnover in the assemblage.

Furthermore, the communities of nitrifying bacteria were affected by acidification (Fig. 6a). The relative abundance of *Nitrosomonas* (the dominant AOB) decreased in the acidified treatments ($P < 0.05$), while

that of *Nitrospira* [the dominant nitrite-oxidizing bacteria (NOB), including comammox] increased ($P < 0.05$). This overall increased ratio of *Nitrospira* over known ammonia oxidizers indicates an increasing proportion of comammox under acidified conditions. Indeed, the comammox *amoA* gene abundance was higher in the acidified treatments ($P < 0.05$) (Supplementary Fig. 16d). In addition, based on metatranscriptome data, higher comammox gene transcripts were detected in the acidified treatments (0.78–11.59% of the total obtained gene transcripts involved in the stepwise oxidation of $NH_3$ to $NO_3^-$) compared with the ambient control (only 0.03–0.89%, $P < 0.05$) (Fig. 6b and Supplementary Fig. 17). Interestingly, the increase was higher for gene transcripts involved in the oxidation of $NH_3$ to $NO_2^-$ (*amo* and *hao*) than those involved in $NO_2^-$ oxidation (*nxr*) (Fig. 6b), suggesting the possibility that part of the comammox may only serve as partial, rather than complete, ammonia oxidizers under acidified conditions. However, further investigations are still required to test

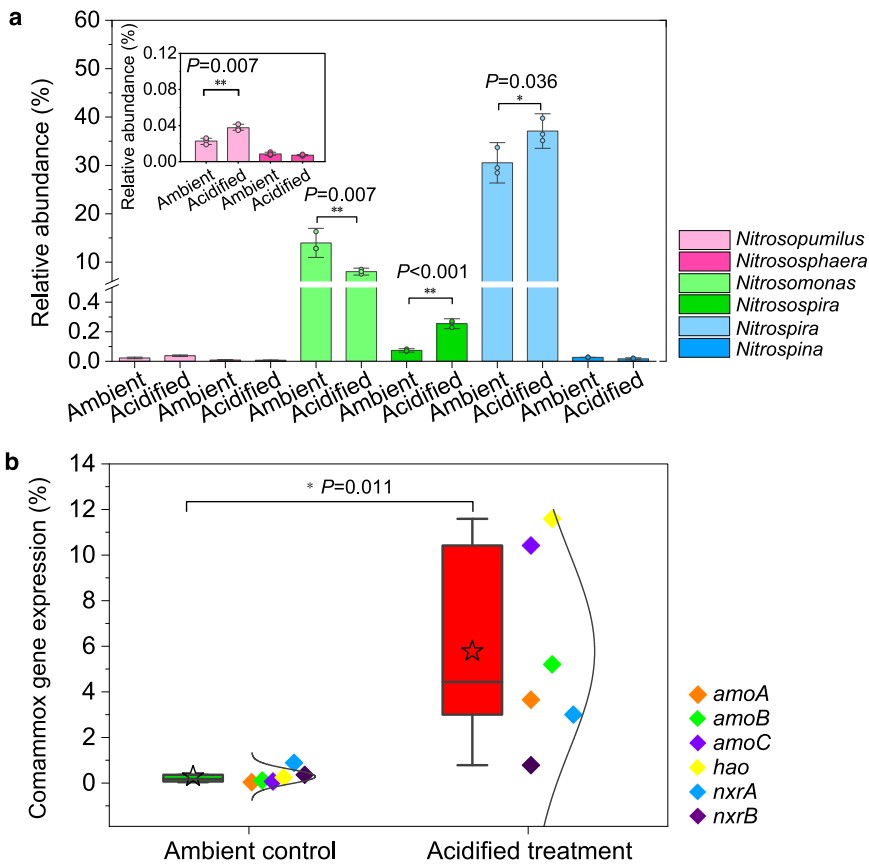

**Fig. 6 | Community composition of nitrifying microbes of the enrichment cultures at the ambient control ($pCO_2$ = 400 μatm; pH = 8.1) and acidified treatments ($pCO_2$ = 800 μatm; pH = 7.8). a** Relative abundance of nitrifying microbes based on the 16S rRNA gene sequencing with universal primers capable of detecting both bacteria and archaea within the same sequencing libraries. The inserted plot shows the relative abundance of ammonia-oxidizing archaea (*Nitrosopumilus* and *Nitrososphaera*). Single asterisk denotes significant difference at $P < 0.05$, while double asterisk denotes significant difference at $P < 0.01$. Error bar represents SD ($n$ = 3 biologically independent samples), and dots are corresponding data points of the replicates. **b** The proportion of key nitrifying gene transcripts of comammox based on metatranscriptome sequencing data. Horizontal lines in the box charts indicate the medium, stars represent the mean. The boxes give the 25th and 75th percentiles, the whiskers show range from 5th to 95th percentiles, and the curves show the distribution of the values. Single asterisk denotes significant difference at $P < 0.05$ ($n$ = 6 genes). Significant differences were determined via one-way analysis of variance (ANOVA). Source data are provided as a source data file.

this hypothesis. These data demonstrate that aquatic acidification has profound impact on nitrifying communities and their physiological metabolism in estuarine and coastal ecosystems.

In conclusion, our findings suggest that the ongoing aquatic acidification due to the synergistic effects of human activity-induced eutrophication and elevated atmospheric $CO_2$ could disrupt a vital link of nitrogen cycle and increase the production of the powerful greenhouse gas $N_2O$ in estuarine and coastal waters. Contrary to our expectation, elevated $pCO_2$ did not exhibit "$CO_2$-fertilization" effect on chemoautotrophic nitrifiers under acidified conditions. Nitrifiers respond significantly to acidification at the transcriptional level, and greatly up-regulate gene expressions associated with intracellular pH homeostasis to cope with acidification stress. These results provide important insights about the underlying mechanism of acidification affecting nitrification and associated $N_2O$ emission, and we propose that further acidification in estuarine and coastal waters may alter nitrogen cycle and accelerate global warming by stimulating $N_2O$ emission.

## Methods

### Study sites and sample collection
The Yangtze River is the largest river in the Euro-Asian continent by virtue of water discharge and is also the third longest river in the world. The Yangtze River basin is characterized by rapid economic development and high population density. Due to the intensive human activities, significant amount of reactive N has been discharged into the Yangtze Estuary and adjacent coastal area, leading to serious N pollution and rapid water acidification[65]. Therefore, the Yangtze Estuary was selected as the study area to investigate the responses of nitrification rate and associated $N_2O$ emission to aquatic acidification. To achieve this goal, six representative sampling sites (Yz1 to Yz6) were selected along the estuarine gradient from the estuary mouth to its adjacent coastal area.

In this study, near-bottom waters, which typically exhibit more active nitrification and more severe acidification[65,66], were collected from these sites during the March cruise of National Natural Science Foundation of China in 2020, using Niskin-X bottles mounted on a conductivity-temperature-depth (CTD) profiler (Sea-Bird 911 plus) (Fig. 1). Water depth, salinity, pH and DO were recorded with CTD profiler, pH meter (Thermo Orion 3-Star) and Winkler's method. Part of the water from each site was preserved in dark at 4 °C for subsequent acidification experiments. Meanwhile, known amounts of subsamples were immediately filtered with 0.22 μm pore-size sterile filters (Waterman), and the filtrates were preserved for nutrient analyses while the membranes were carefully preserved under −20 °C for later DNA extraction (Supplementary Methods). An extra amount of subsample from site Yz3 was also preserved at 4 °C in dark for later long-term manipulation experiments.

## Acidification manipulation and measurements of nitrification and N₂O production rates

The acidification experiments were conducted in a series of continuous-flow manipulation systems (constructed based on bioreactors, Infors, Switzerland, with 4.0 L water samples and 1.0 L headspace) by continuously and gently bubbling water samples with humidified and 0.22 µm-filtered synthetic air (79% $N_2$ and 21% $O_2$):$CO_2$ mixtures (20 mL min⁻¹, precisely controlled by mass flow meter). The inlet air:$CO_2$ bubbles were gently dispersed in the water body by two stirrers which were installed at the bottom and above the inlet airflow (-30 rpm), to make the water system homogeneous and to equilibrate the liquid and gas phases. After equilibrium, headspace gas was collected from the reactors using gas-tight syringes for analyses of natural isotopic signatures of N₂O to reveal the N₂O production pathways (Supplementary Methods). Subsequently, nitrification rates were determined by addition of $^{15}NH_4Cl$ (>98 atom percent $^{15}N$, lower than 20% of ambient concentration) as a tracer and the accumulation of $^{15}N$ in the $NO_x^-$ ($NO_3^- + NO_2^-$) pools. During rate measurements, water pH was maintained stable via 0.2 M NaOH solution through the reactor's acid–base automatic regulator, as protons ($H^+$) can be released during the process of $NH_3$ oxidation. Meanwhile, DO concentration was maintained saturated and $p$CO₂ was maintained at the targeted level via a 100 mL h⁻¹ gas flow, and the outflow gas was collected with gas sampling bags (Teflon®FEP, DuPont). Temperature was maintained at room temperature (25 °C) by an automatic heating plate and cold circulating water bath. During the incubation, pH, DO, and temperature were recorded in real time through the equipped pH sensor (Hamilton, Switzerland), DO sensor (Hamilton, Switzerland), and temperature sensor (Infors, Switzerland), respectively. The incubations were conducted in dark by covering the reactor tanks with opaque paper. Liquid samples (30 mL) were collected at the beginning and end of the 24 h incubation, with gas-tight syringes through a reserved sampling tube (clamped tightly after sampling) and filtered immediately (0.22 µm, Waterman). Part of the filtered water was used for measurements of $NO_3^-$ and $NO_2^-$, while the other part was prepared for $^{15}NO_x^-$ analysis using the "denitrifier method"[67]. In addition, another 30 mL water samples were collected for measurements of DIC, alkalinity, and $p$CO₂. Meanwhile, gas samples were extracted using gas-tight syringes for $CO_2$, N₂O, and N₂O isotope measurements. Before utilization, the reagent solutions were filter-sterilized (0.22 µm, Waterman), and the reactor tank and related materials were heat-sterilized (121 °C and 15 psi for 20 min) (the same below). All the experiments were conducted in triplicate. The nitrification rates were calculated using the following equation[68]:

$$R_{nitrification} = \frac{\left(R_t NO_X^- \times [NO_X^-]_t\right) - \left(R_{t0} NO_X^- \times [NO_X^-]_{t0}\right)}{t - t0} \times \frac{1}{F} \tag{1}$$

$$F = \frac{[^{15}NH_4^+]}{[^{14}NH_4^+] + [^{15}NH_4^+]} \tag{2}$$

where $R_{nitrification}$ is the nitrification rate (nmol L⁻¹ h⁻¹), $R_{t0}NO_X^-$ and $R_t NO_X^-$ are the ratios of $^{15}N$ in the $NO_X^-$ pool measured at the initial ($t0$) and final ($t$) time of the incubation, respectively. $[NO_X^-]_{t0}$ and $[NO_X^-]_t$ are the concentrations of $NO_X^-$ at the initial ($t0$) and final ($t$) time of the incubation, respectively. $[^{14}NH_4^+]$ and $[^{15}NH_4^+]$ represent the ambient $NH_4^+$ concentration and the final $^{15}NH_4^+$ concentration after the addition of the stable isotope tracer, respectively.

N₂O production rates were calculated based on the increase in mass 45 and 46 N₂O ($^{45}N_2$ and $^{46}N_2$) during the experiments[69]:

$$R_{N_2O} = \frac{1}{F} \times \left( \frac{^{45}N_2O_t - {^{45}N_2O_{t0}}}{t - t0} + 2 \times \frac{^{46}N_2O_t - {^{46}N_2O_{t0}}}{t - t0} \times \frac{1}{F} \right) \times \frac{1}{V} \tag{3}$$

where $R_{N_2O}$ is the N₂O production rate (pmol N₂O·N L⁻¹ h⁻¹). $F$ is the fraction of $^{15}N$ in the substrate ($NH_4^+$) pool, as described above. $^{45}N_2O_{t0}$ and $^{45}N_2O_t$ indicate the amount of $^{45}N_2O$ at the initial ($t0$) and final ($t$) time of the incubation, respectively. $^{46}N_2O_{t0}$ and $^{46}N_2O_t$ are the amount of $^{46}N_2O$ at the initial ($t0$) and final ($t$) time of the incubation, respectively. $V$ represents the volume of water sample (L).

## Manipulation experiments to decouple the effects of elevated $p$CO₂ and reduced pH

To distinguish potential effects of increasing $p$CO₂ and decreasing pH on nitrification rates and associated N₂O production under acidification, a series of continuous-airflow manipulation systems were constructed similarly as in the acidification experiments. Briefly, four groups of simulation systems were constructed: (a) 400 µatm/pH 8.1 (the ambient control group), (b) 400 µatm/pH 7.8 (reduction of pH only, maintaining $p$CO₂ at the ambient level), (c) 800 µatm/pH 7.8 (the acidification group), (d) 800 µatm/pH 8.1 (increase of $p$CO₂ only, maintaining pH at the ambient level) (Supplementary Fig. 7). 4 L of the collected water samples from site Yz3 was added per reactor, and $^{15}NH_4Cl$ (>98 atom percent $^{15}N$) was added to lower than 20% of ambient $NH_4^+$ concentrations. The carbonate chemistry was manipulated by steadily bubbling the water samples with 0.22 µm-filtered $CO_2$ adjusted air (400 µatm for groups a and b, 800 µatm for groups c and d) while adjusting pH (7.8 for groups b and c, 8.1 for groups a and d) with sterile acid (0.1 M HCl) or base (0.2 M NaOH) solution via the reactor's acid-base automatic regulator. Other reaction conditions (temperature, DO, gas flow rate, stirring speed, and dark condition) were the same as in the acidification experiments. Nitrification rates and N₂O production rates were measured as described above during 24 h of incubation after equilibrium. All the experiments were conducted in triplicate.

## Long-term acidification manipulation

Another two groups of manipulation systems were set up similarly with water samples from site Yz3 for long-term acidification experiments: (a) 400 µatm/pH 8.1 (control group) and (b) 800 µatm/pH 7.8 (acidification group). $p$CO₂ and pH in the water body were manipulated by continuous bubbling with humidified and 0.22 µm-filtered ambient air (400 µatm, group a) or $CO_2$-enriched air (800 µatm, group b), with pH stabilized at 7.8 or 8.1, respectively. During the long-term acidification experiments, $NH_4^+$ was supplemented and maintained at around ambient concentrations (below 10 µM) in the reactors (Supplementary Fig. 10). After about half of the $NH_4^+$ was consumed, filter-sterilized site water samples with proper $NH_4^+$ concentrations were used as culture medium and supplied at a flow rate of 1 L day⁻¹ to all reactors. Sterilized NaOH solution of 0.2 M was used to neutralize the released $H^+$ in the process of nitrification. The DO was maintained saturated and the temperature was maintained at 25 °C as described above. During the incubation, liquid samples (10 ml) were collected from the reactors every day and immediately filtered (0.22 µm, Waterman) for measurements of $NH_4^+$, $NO_3^-$, and $NO_2^-$. 2 ml of the headspace gas was also collected every day using gas-tight syringes for analyses of N₂O and $CO_2$. Known amount of sample was collected every several days and pelleted by centrifugation (20,000 g, 5 min). Pellets were immediately used for DNA extraction, and subsequent pyrosequencing and qPCR assays (Supplementary Methods, Supplementary Table 10). Subsamples were also collected every week for measurements of nitrification and N₂O production rates. At the end of one-month incubation, samples were harvested, pelleted (20,000 g for 5 min, under 2 °C), and cryopreserved immediately in liquid nitrogen for subsequent metatranscriptomic analyses.

## Enrichment of ammonia-oxidizing archaea (AOA)

For AOA enrichment, a continuous-flow nitrifying bioreactor (Infors, Switzerland) with a working volume of 4.0 L was set up with fresh water

samples from site Yz3. Filter-sterilized site water supplemented with $NH_4^+$ (~10 μM) and streptomycin (~50 mg/L, to inhibit nitrifying bacteria[70,71]) was continuously supplied at a flow rate of ~0.5 L day$^{-1}$. DO concentration was maintained saturated by flushing continuously with air. pH was maintained at 8.1 with 0.2 M $KHCO_3$ solution through the reactor's acid-base automatic regulator. The incubations were conducted in dark and the temperature was maintained at 25 °C as described above. During the incubation, liquid samples were collected from the reactor and filtered immediately using 0.22 μm pore size filters (Waterman) for the measurement of $NH_4^+$, $NO_3^-$, and $NO_2^-$. In addition, samples were collected (20,000 $g$, 5 min) for DNA extraction and subsequent pyrosequencing to monitor the enrichment of AOA (Supplementary Methods). After 50 days of incubation, AOA enrichment culture was harvested to measure nitrification and $N_2O$ emission rates.

### Measurements of nitrification and $N_2O$ emission rates of the enrichment culture

Liquid samples collected from the reactors were centrifuged (6,000 $g$, 5 min) to harvest the nitrifying-enriched biomass[27]. Then, the precipitated biomass at the bottom of the centrifuge tube was washed three times using filter-sterilized site water and re-suspended in it before rate measurements. Subsequently, 100 μl of this suspension was transferred into gas-tight glass vials (120 ml) in which 10 ml of fresh filter-sterilized site water with $NH_4^+$ (~10 μM final concentration) was sufficiently bubbled with $CO_2$ adjusted air (400 μatm or 800 μatm). During the incubation, pH was maintained at the targeted level via addition of 3-(N-morpholino) propanesulfonic acid (MOPS, pH adjusted to 8.1 or 7.8, 20 mM final concentration)[23]. The vials were incubated in dark at 25 °C on an orbital shaker (30 rpm). At each sampling interval (0, 5, 8, 12, and 24 h), three replicates were sacrificed and 2 ml of the headspace gas was extracted for $CO_2$ and $N_2O$ analyses. Targeted pH and DO were confirmed using a Mettler-Toledo pH Meter and an OXY Meter S/N 4164 with an oxygen needle sensor (Unisense), respectively. Meanwhile, 5 ml of suspension in each vial was immediately filtered (0.22 μm, Waterman) for $NO_x^-$ measurements. Nitrification rates were estimated on the basis of the linear changes in $NO_x^-$ concentrations with time[23]. Inhibition of nitrification activity was expressed as the percentage reduction of nitrification rate in the acidified treatments compared to the ambient control. Effect of acidification on $N_2O$ emission was determined based on the percentage change of $N_2O$ concentration (in the headspace of the glass vials) in the acidified treatments relative to the ambient control[23].

### RNA extraction and metatranscriptomic analyses

Total RNA was extracted from triplicate ambient controls (400 μatm/pH 8.1) and acidified treatments (800 μatm/pH 7.8) using EZNA® Soil RNA kit (Omega Bio-tek, Norcross, GA, USA) at the end of incubation. Residual genomic DNA was removed with the Turbo DNA-free kit (Ambion) and further verified through PCR using primers 515F (5′-GTGCCAGCMGCCGCGGTAA-3′) and 909R (5′-CCCCGYCAATTCMTTT RAGT-3′) to rule out DNA contamination[23]. The purity, integrity, and concentration of the extracted RNA were measured using Agilent2100 (Agilent) and Nanodrop2000 (Thermo). Ribosomal RNA (rRNA) was then removed via Ribo-Zero rRNA Removal Kit (Epicentre) to acquire qualified mRNA, which was used for qPCR analyses (Supplementary Methods) and metatranscriptome sequencing.

Metatranscriptomic cDNA libraries were constructed with the TruSeq RNA Sample Prep Kit (Illumina) and sequenced on an Illumina HiSeq4000 platform after triplicate mRNA samples from the ambient controls and the acidified treatments were pooled individually[72]. The quality of raw reads was checked via FastQC and trimmed by SeqPrep (https://github.com/jstjohn/SeqPrep). Subsequently, raw reads were quality-filtered using Sickle (https://github.com/najoshi/sickle), and sequences with ambiguous (N) bases, low quality (below 20), and

lengths less than 50 base pairs (bp) were discarded. SortMeRNA (http://bioinfo.lifl.fr/RNA/sortmerna/) was further used to screen and remove rRNA reads. The resulting high-quality mRNA clean reads were assembled using Trinity de novo assembly pipeline (contigs less than 200 bp were removed)[73]. MetaGeneMark (http://exon.gatech.edu/meta_gmhmmp.cgi) was used to predict open reading frames (ORFs), and those longer than 100 bp were translated into amino acid sequences. Non-redundant contigs (95% identity; 90% coverage) were obtained via CD-HIT software (http://www.bioinformatics.org/cd-hit/). Expression levels of the transcripts were calculated via kallisto (https://pachterlab.github.io/kallisto/) and were reported as TPM (Transcripts Per Kilobase Million). Fold change (FC) in relative gene expression was calculated by comparing the acidified treatments to the ambient control. Taxonomic affiliations of the transcripts were assigned via binning to the best hit in the NR database (BLASTP, $e$-value ≤ $10^{-5}$). Potential functions were assigned based on the best homology to proteins within the KEGG (Kyoto Encyclopedia of Genes and Genomes) database (BLASTP, $e$-value ≤ $10^{-5}$). Maximum-likelihood tree was constructed with IQ-TREE[74] with 1000 ultrafast bootstraps and then visualized and annotated by iTOL (interactive tree of life)[75].

### Analytical methods

Concentrations of $CO_2$ and $N_2O$ were monitored by gas chromatography (GC-2014, Shimadzu, Kyoto, Japan). $N_2O$ isotope ratios ($m/z$ = 44, 45, 46) were analyzed on isotope ratio mass spectrometry (IRMS, Delta V Advantage, Thermo Fisher Scientific, Bremen, Germany). Natural $N_2O$ isotopic signatures ($\delta^{15}N^{bulk}$, $\delta^{15}N^{\alpha}$, and $\delta^{18}O$) for revealing $N_2O$ production pathways were analyzed on IRMS (Delta V Plus, Thermo Fisher Scientific, Bremen, Germany). Concentrations of $NH_4^+$, $NO_3^-$ and $NO_2^-$ were measured colorimetrically using a continuous-flow nutrient analyzer (Skalar SANplus, Skalar Analytical BV, Breda, The Netherlands) with detection limits of 0.3 μM for $NH_4^+$-N and 0.05 μM for $NO_2^-$-N and $NO_3^-$-N. DIC concentration was analyzed by acidification and subsequent quantification of released $CO_2$ (Carbon coulometer, UIC-INC, America). Alkalinity and $pCO_2$ were calculated via the CO2SYS program[76], on the basis of pH and DIC measurements using the carbonic acid dissociation constants of ref. [77] that were refit in different functional forms by ref. [78].

### Statistical analyses

Statistical analyses were performed using SPSS version 19.0 for Windows (SPSS Inc., Chicago, I L. USA). Significant differences among differently treated groups were identified via one-way analysis of variance (ANOVA) followed by Tukey's honestly test. Nonlinear fitted curves (polynomial fit) were constructed using Origin 2022b to explore the responses of nitrification and associated $N_2O$ production rates to different levels of acidification. Results were considered significant when $P < 0.05$.

### Reporting summary

Further information on research design is available in the Nature Portfolio Reporting Summary linked to this article.

## Data availability

All sequence data and sample information are available at National Center for Biotechnology Information (NCBI) Sequence Read Archive (SRA) database under BioProject accession numbers PRJNA876082. All data needed to evaluate the conclusions in the paper are present in the paper and/or the Supplementary Materials. Source data are provided with this paper.

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

## Acknowledgements

This work is supported by the National Natural Science Foundation of China (Nos. 41725002 [L.J.H.], 41971105 [Y.L.Z.], 42222605 [Y.L.Z.], 42030411 [L.J.H.], 42249903 [L.J.H.], 41671463 [L.J.H.], and 41730646 [M.L.]), the Chinese National Key Programs for Fundamental Research and Development (No. 2016YFA0600904) [L.J.H.], and Director's Fund of Key Laboratory of Geographic Information Science (Ministry of Education), East China Normal University (Grant No. KLGIS2022C03) [Y.L.Z.]. Samples were collected during the open research cruises to Yangtze Estuary (NORC2020-03 and NORC2023-302). We thank W. S. Gardner and Silvia E. Newell for discussion and revision on the manuscript.

## Author contributions

Y.L.Z., L.J.H., and M.L. conceived the research. J.Z., Y.L.Z., L.J.H., Z.R.A., F.Y.C., B.L.L., L.W., and L.Q. performed the research. J.Z., Y.L.Z., L.J.H., H.P.D., P.H., G.Y.Y., X.L., Y.Y., X.F.L., D.Z.G., Y.L., Z.F.L., R.B., and M.L. analyzed the data. J.Z., Y.L.Z., L.J.H., R.B., and M.L. wrote the paper.

## Competing interests

The authors declare no competing interests.
