## [Peer Review File · Nature Communications]

Effects of acidification on nitrification and associated nitrous oxide emission in estuarine and coastal watersReviewer #1 (Remarks to the Author):

The topic of the manuscript "Effects of acidification on nitrification and associated nitrous oxide emission in estuarine and coastal waters" by Zhou et al. is very interesting and worth study. Lots of tests were conducted and experimental data were obtained. The authors carried out analysis and mining of these experimental data and then gotten interesting results. However, theoretical description is not enough and the evidence is also insufficient. So, I think, the manuscript can only be accepted after some major revisions.

1] As described as "the measured inhibition of nitrification rate by acidification in the estuarine and coastal waters was generally lower than that in the oligotrophic open oceans where nitrification rates were reported to decline by 3-44% in response to a decrease of 0.1 pH unit", was here the differences in the natures of Nutrients among the water samples collected " 6 representative sites (Yz1 to Yz6)" ? Did these natures have influences on the nitrification rate? To answer these issues, therefore, more composites of the samples should be given in the paper. Especially, "oligotrophic open oceans", "toxicant exposures", and "human activity-induced eutrophication" were employed to elucidate the influences of acidification on nitrification and its effects.

2] it is good that "nitrifying communities" " different nitrifying organisms" and even "heterotrophic denitrifiers" were employed to explain the pathway of N₂O production. Even so, more evidences and data were wanted to prove that "the contribution of denitrifying bacteria to the production of N₂O should be negligible."

3] to research "Decoupling potential effects of elevated pCO₂ and reduced pH", the methods including adjusting pH (7.8 and 8.1) with sterile acid or base solution were used as Breider et al.³⁷ did. In the present study, what had been done to avoid changing alkalinity and carbonate parameters of the studied water ?

4] "previous attempts to acquire metatranscriptomes based on the short-term acidification experiments failed, because there was no enough qualified mRNA with good integrity and purity, which might be due to the extreme instability of mRNA and the complexity of environmental samples¹⁸". I think, these are just appearances, but the essence is as you described that " nitrifiers account for only a small fraction of the complex microbial communities in estuarine and coastal waters ". So, it is good that " a series of continuous-flow environmental simulators with water samples from site Yz3 were set up to mimic long-term acidification and to enrich nitrifiers."

5] when these experimental data Data analysis and mining were done, more comparison studies can be done with the results of other researchers.

6] Line 267, what are short for "amoB (also suggested as a catalytic subunit⁴⁷ 267) and amoc" ?

7] Line 403, should " word" be replaced by "world"?

Reviewer #2 (Remarks to the Author):

"Effects of acidification on nitrification and associated nitrous oxide emission in estuarine and coastal waters" By Zhou et al

This manuscript reports on an intensive experimental investigation into the effect of ocean acidification on nitrification rates and nitrous oxide production in estuarine and coastal waters. A particular and noteworthy factor of this work relative to previous investigations is the detailed analysis of the metatranscriptomics associated with these processes. Whilst this manuscript is thorough in detail and the results are novel and of importance to our understanding of contemporary and future nitrogen biogeochemistry I find that the current presentation is far too reliant on supplementary material, requiring the reader to constantly change between the two documents. This current organisation renders this paper hard to follow and difficult to interpret. I did not even realise that there had been measurements of nitrification and nitrous oxide in an

enriched culture additional to the unenriched experiment until reaching the end of my review, and whilst this enrichment experiment supports the overall findings there is no discussion about the relevance or potential experimental artefact offered by this treatment. My overriding suggestion is not to remove the supplementary material, but to integrate it better into the main paper and to re-organise so that the whole document is easier for the reader to follow.

Whilst there is a fairly comprehensive review and integration of previous literature, I would suggest that the work of Fulweiler et al (2011 - DOI 10.1007/s12237-011-9432-4) and Rees et al (2016 -<http://dx.doi.org/10.1016/j.dsr2.2015.12.006>) should both be incorporated into discussions here. Fulweiler's work showed that for estuarine/coastal conditions there was an inverse relationship between pH and nitrification, that is the opposite effect to that shown here. Whilst Rees showed that N₂O concentration decreased with increasing OA and saw no change in numbers of AOA both of which are different to here. The study of Rees was in open water and in very different conditions to this paper, but as one of very few papers on this subject probably offers some discussion points pertinent to the control of N₂O release from nitrification.

Specific comments:

L33 Whilst offering important insights the claim that this work will "help predict climate change" is taking things too far.

L121-122 eg Fulweiler et al

L129 disturbance on nitrification COULD cause profound consequences

L133 Beman predicted, Rees et al confirmed for open ocean conditions

L234-235 There was NOT enough mRNA with good integrity

L245 I'm not convinced that the data shown in Supplementary Fig 8 shows that steady state was reached after 25 days. There needs to be some informed discussion on this figure. I can't see why ammonium concentration in "Influent" was as high as 60µM, the ambient at Yz3 was ~10µM, but then where does it go, as is close to zero in effluent – surely not all is nitrified??

L218-224. This is hard to follow and needs re-writing

L226 Transcriptional response of nitrifiers to acidification during long-term incubation

L278 Supplementary Fig 15 referred to before 12, 13, 14

L564 nutrient analysis likely to be colorimetrically rather than spectrophotometrically
Fig 3 How do "different lowercase letters above the columns indicate significant differences"?

Supp Fig 9 b – Typo Environment

Supp Fig 14 & 16 – no reference to this figure found in text

Supp Table 9, 10, 11 – no reference to this table found

Reviewer #3 (Remarks to the Author):

This paper addresses an important issue relating to climate change and environmental impacts in the coastal zone. Authors used a suite of technologies to investigate nitrification and emission of nitrous oxides under ocean acidification, overall the paper has potential in getting published in Nature Communications, but have the following concerns.

Major comments:

1. Although the authors distinguished the effects of acidification from pCO₂ and pH only, acidification decreased nitrification rate and increased N₂O production. This main result has been shown in a previous study (<https://doi.org/10.1038/s41558-019-0605-7>).

2. I was confused by the sampling strategy. It seems that the bottom (sedimentary) water was taken? why not sample the surface water that is mainly affected by acidification. Also, it is widely accepted that nitrification functions well in the surface,

and it is less likely in the bottom water.

3. Why not consider Comammox process, not only on transcript level? I think this is important when evaluating nitrification process. Did authors conduct amplicon sequencing?

Minor comments:

1. what does "CO₂-fertilization" mean? This is mainly used to describe this effect in terrestrial ecosystem. It should be avoided using any ambiguous words in abstract;

2. what does "efficient changes" in gene expression refer to in the abstract? This sentence here does not mean much sense, it does not convey any practical info;

3. Misleading statement. "However, how microbially-mediated biogeochemical processes may be altered by acidification in estuarine and coastal ecosystems remains largely unknown, to a great extent limiting our understanding of the responses of these ecosystems to global environmental changes". Many studies have explored the microbially-mediated biogeochemical processes in estuarine and coastal ecosystems. E.g., DOI10.1038/s41561-020-0584-3; DOI10.1021/acs.est.2c00692; DOI10.1002/2014GL060849; DOI10.1016/j.scitotenv.2020.142689; DOI10.1038/s41467-017-00417-7

4. Line 95, delete "new";

5. Line 377 Indeed, the comammox amoA gene abundance was higher in the acidified treatments, how did the authors confirm that this amoA gene is affiliated to comammox, not for AOA or AOB?

Overall, I think is work is interesting and authors should improve the MS to distill the major significance better, as opposed to previously reported results, as indicated earlier.

Point-by-point response to the reviewers' comments

Reviewer #1 (Remarks to the Author):

Comment: The topic of the manuscript “Effects of acidification on nitrification and associated nitrous oxide emission in estuarine and coastal waters” by Zhou et al. is very interesting and worth study. Lots of tests were conducted and experimental data were obtained. The authors carried out analysis and mining of these experimental data and then gotten interesting results. However, theoretical description is not enough and the evidence is also insufficient. So, I think, the manuscript can only be accepted after some major revisions.

Response: Thank you very much for your comments. This manuscript has been carefully modified according to your suggestions, and all the raised issues have been addressed as follows.

Comment: 1] As described as “the measured inhibition of nitrification rate by acidification in the estuarine and coastal waters was generally lower than that in the oligotrophic open oceans where nitrification rates were reported to decline by 3-44% in response to a decrease of 0.1 pH unit”, was here the differences in the natures of Nutrients among the water samples collected “6 representative sites (Yz1 to Yz6)”? Did these natures have influences on the nitrification rate? To answer these issues, therefore, more composites of the samples should be given in the paper. Especially, “oligotrophic open oceans”, “toxicant exposures”, and “human activity-induced eutrophication” were employed to elucidate the influences of acidification on nitrification and its effects.

Response: In this study, we determined the concentrations of nitrogen nutrients in all water samples collected from sites Yz1 to Yz6, which showed a decreasing trend from the upper estuary to the coastal area, due to the watershed anthropogenic nutrient inputs. This difference leads to higher nitrification rate in the upper estuary waters (please see Supplementary Fig. 2) and also affects the response of nitrifiers to acidification along the nutrient gradient. In the revised manuscript, the correlation between NH_4^+ concentration and the effects of acidification on nitrification rates has been supplemented based on this study and the previously reported data. Results showed a significant negative relationship between NH_4^+ concentration and the inhibition effect of acidification on nitrification rate (please see Supplementary Fig. 4). In addition, environmental parameters of the water samples are provided in Supplementary Table 1, and the following information has also been provided in the revised manuscript:

“Especially, the relatively higher NH_4^+ concentrations in the upper estuaries may mitigate the inhibiting effects on ammonia oxidation caused by acidification (Supplementary Table 1). Consistently, the measured inhibition of nitrification rate by acidification in the estuarine and coastal waters was generally lower than that in the oligotrophic seas where nitrification rates were reported to decline by 3-44% in response to a decrease of 0.1 pH unit^{28,42}. Indeed, NH_4^+ concentration was negatively correlated with the inhibition effect of acidification on nitrification rate in different habitats ranging from estuary to open ocean ($P < 0.05$)^{28,42,44} (Supplementary Fig. 4)” (Please see lines 124-132).

Comment: 2] it is good that “nitrifying communities” “different nitrifying organisms” and even “heterotrophic denitrifiers” were employed to explain the pathway of N_2O production. Even so, more evidences and data were wanted to prove that “the contribution of denitrifying bacteria to the production of N_2O should be negligible.”

Response: To explore the pathway of N_2O production, ^{15}N -site preference (SP) value was calculated by the isotopomer ratios as shown below:

$$\delta^{15}\text{N}^{\text{bulk}} = (\delta^{15}\text{N}^{\alpha} + \delta^{15}\text{N}^{\beta})/2$$

$$SP = \delta^{15}\text{N}^{\alpha} - \delta^{15}\text{N}^{\beta}$$

Characteristic SP values of 33‰ for NH_2OH oxidation and 0‰ for NO_2^- reduction, which were estimated in pure cultures (Sutka et al., 2006; Rathnayake et al., 2013), were used to estimate the contribution of each process, assuming that each process is linearly proportional to the SP value using the following equations (Rathnayake et al., 2013).

$$F_{\text{NH}_2\text{OH oxidation}}(\%) = \frac{(SP - SP_{\text{NO}_2^- \text{ reduction}})}{(SP_{\text{NH}_2\text{OH oxidation}} - SP_{\text{NO}_2^- \text{ reduction}})} \times 100\%$$

$$F_{\text{NO}_2^- \text{ reduction}}(\%) = 100\% - F_{\text{NH}_2\text{OH oxidation}}$$

where $F_{\text{NH}_2\text{OH oxidation}}$ and $F_{\text{NO}_2^- \text{ reduction}}$ denote the contribution of NH_2OH oxidation and NO_2^- reduction to the total N_2O production.

On this basis, the calculated SP values in our continuous-flow manipulation systems were from 28.6 to 33.8. Thus, it was estimated that 86.7-100% of the released N_2O was produced via NH_2OH oxidation, whereas the contribution of NO_2^- reduction was minor or nil. It should be noted that, the N_2O isotopomer analysis could not distinguish the relative contributions of nitrifier denitrification and heterotrophic denitrification. Although heterotrophic denitrifiers may also contribute to the production of N_2O , their contribution may be insignificant as the samples from all study sites were well oxygenated (DO : 8.30-9.86 mg L^{-1}) and remained at near *in situ* high DO levels during the incubation. Actually, previous studies reported that when the dissolved O_2

concentration is more than 0.06 mg L^{-1} , the N_2O production by denitrification is completely inhibited (Dalsgaard et al. 2014). Therefore, the contribution of heterotrophic denitrifying bacteria to the production of N_2O should be negligible. This point has been clarified in the revised manuscript: “*Although heterotrophic denitrifiers may also contribute to the production of N_2O , their contribution may be insignificant, as the natural isotopic signatures of N_2O showed that the pathway of NO_2^- reduction (including nitrifier denitrification and heterotrophic denitrification) contributed only 0-13.3% of the released N_2O (Supplementary Table 4). Moreover, the samples from all study sites were well oxygenated [dissolved oxygen (DO): $8.30\text{-}9.86 \text{ mg L}^{-1}$; Supplementary Table 1] and remained at high DO levels during the incubation (Supplementary Table 2), which was unlikely to occur for heterotrophic denitrification. Previous studies reported that when the DO concentration is more than 0.06 mg L^{-1} , the N_2O production by heterotrophic denitrification is completely inhibited⁴⁹. Therefore, the contribution of denitrifying bacteria to the production of N_2O should be negligible*” (Please see lines 174-184); “*headspace gas was collected from the reactors using gas-tight syringes for analyses of natural isotopic signatures of N_2O to reveal the N_2O production pathways (Supplementary Methods)*” (Please see lines 456-458).

Comment: 3] to research “*Decoupling potential effects of elevated $p\text{CO}_2$ and reduced pH*”, the methods including adjusting pH (7.8 and 8.1) with sterile acid or base solution were used as Breider et al. 37 did. In the present study, what had been done to avoid changing alkalinity and carbonate parameters of the studied water?

Response: In the study of Breider et al. (2019), acidification was manipulated by adding strong acid (HCl) solution in a capped system. Although the addition of HCl can elevate $p\text{CO}_2$ and reduce pH, it also alters alkalinity and results in different carbonate parameters compared with those expected in the future, i.e., dissolved inorganic carbon (DIC) increases under natural aquatic acidification rather than remains unchanged. In the present study, the acidification experiments were conducted in a series of continuous-flow manipulation systems by continuously and gently bubbling water samples with $0.22 \text{ }\mu\text{m}$ -filtered different air: CO_2 mixtures. This bubbling method increases DIC at constant total alkalinity, which can best mimic the ongoing aquatic acidification. On this basis, to further decouple the potential effects of elevated $p\text{CO}_2$ and reduced pH, the carbonate chemistry was manipulated by steadily bubbling the water samples with $0.22 \text{ }\mu\text{m}$ -filtered CO_2 adjusted air while adjusting pH with sterile acid or base solution via the reactor’s acid-base automatic regulator. These details are all provided in the revised manuscript: “*To distinguish the individual effects of elevated $p\text{CO}_2$ and reduced pH, a series of open, continuous-flow microcosm systems were*

constructed. The carbonate chemistry was manipulated by steadily bubbling collected water samples from site Yz3 with CO₂ adjusted air (400 μ atm and 800 μ atm) while adjusting pH (7.8 and 8.1) with sterile acid or base solution according to a real-time pH detector (Supplementary Fig. 7)” (Please see lines 198-203).

Comment: 4] “previous attempts to acquire metatranscriptomes based on the short-term acidification experiments failed, because there was no enough qualified mRNA with good integrity and purity, which might be due to the extreme instability of mRNA and the complexity of environmental samples”. I think, these are just appearances, but the essence is as you described that “nitrifiers account for only a small fraction of the complex microbial communities in estuarine and coastal waters”. So, it is good that “a series of continuous-flow environmental simulators with water samples from site Yz3 were set up to mimic long-term acidification and to enrich nitrifiers.”

Response: As suggested, the essence has been emphasized in the revised manuscript: “More importantly, nitrifiers account for only a small fraction of the complex microbial communities in estuarine and coastal waters (<5%; Supplementary Fig. 9), thus it is difficult to extract sufficient nitrifying transcripts to fully reveal the physiological metabolism of nitrifiers. Therefore, a series of continuous-flow environmental simulators with water samples from site Yz3 were set up to mimic long-term acidification and to enrich nitrifiers” (Please see lines 251-256).

Comment: 5] when these experimental data analysis and mining were done, more comparison studies can be done with the results of other researchers.

Response: As suggested, more comparison analysis has been supplemented in the revised manuscript: “This inhibition effect of acidification on nitrification rate is consistent with what was previously observed in the open oceans^{28,42}. Whilst nitrification rate was reported to increase along a decreasing natural gradient of pH in Narragansett Bay⁴³, it was likely due to a combination of biogeochemical conditions rather than the effect of acidification alone” (Please see lines 114-118); “Consistently, the measured inhibition of nitrification rate by acidification in the estuarine and coastal waters was generally lower than that in the oligotrophic seas where nitrification rates were reported to decline by 3-44% in response to a decrease of 0.1 pH unit^{28,42}. Indeed, NH₄⁺ concentration was negatively correlated with the inhibition effect of acidification on nitrification rate in different habitats ranging from estuary to open ocean ($P<0.05$)^{28,42,44} (Supplementary Fig. 4)” (Please see lines 127-132); “The increased N₂O production under acidified conditions in estuarine and coastal waters is consistent with those in the western North Pacific⁴². However, Rees et al.⁴⁴ documented inhibition

of N₂O production by ocean acidification in cold temperate and polar seawaters. Assuming that nitrification is the main N₂O production pathway in their study, the response of the N₂O production to acidification would be different in polar seas” (Please see lines 169-174).

Comment: 6] Line 267, what are short for “*amoB* (also suggested as a catalytic subunit) and *amoC*”?

Response: As suggested, the explanations of *amoB* and *amoC* have been provided in the revised manuscript: “expressions of bacterial *amoB* (ammonia monooxygenase subunit B, also suggested as a catalytic subunit⁵⁶) and *amoC* (ammonia monooxygenase subunit C) decreased by 68% and 69%, respectively, after acidification” (Please see lines 284-286).

Comment: 7] Line 403, should “word” be replaced by “world”?

Response: It has been revised in the manuscript (Please see line 425).

Reviewer #2 (Remarks to the Author):

Comment: “Effects of acidification on nitrification and associated nitrous oxide emission in estuarine and coastal waters” By Zhou et al. This manuscript reports on an intensive experimental investigation into the effect of ocean acidification on nitrification rates and nitrous oxide production in estuarine and coastal waters. A particular and noteworthy factor of this work relative to previous investigations is the detailed analysis of the metatranscriptomics associated with these processes. Whilst this manuscript is thorough in detail and the results are novel and of importance to our understanding of contemporary and future nitrogen biogeochemistry I find that the current presentation is far too reliant on supplementary material, requiring the reader to constantly change between the two documents. This current organisation renders this paper hard to follow and difficult to interpret. I did not even realise that there had been measurements of nitrification and nitrous oxide in an enriched culture additional to the unenriched experiment until reaching the end of my review, and whilst this enrichment experiment supports the overall findings there is no discussion about the relevance or potential experimental artefact offered by this treatment. My overriding suggestion is not to remove the supplementary material, but to integrate it better into the main paper and to re-organise so that the whole document is easier for the reader to follow.

Response: Thank you very much for your comments and suggestions. As suggested, we have incorporated necessary supplementary materials into the main paper to make

the whole manuscript easier to follow. Especially, measurements of nitrification and nitrous oxide in the enriched culture have been provided in the revised “Material and methods” of the main text (Please see lines 560-580), and in the revised “Results and discussion”: *“For these nitrifying enrichment cultures, significant reduced nitrification rates and stimulated N₂O emissions were observed in the acidified treatments (Supplementary Figs. 14 and 15), a pattern consistent with that of the field water samples”* (Please see lines 268-271). In addition, discussions about the experimental artefact offered by this enrichment experiment have also been supplemented: *“Although the enrichment manipulation changes the original environmental microbial communities, the nitrifying enrichments are representatives to investigate the transcriptional response of nitrifiers to acidification in complex estuarine and coastal waters”* (Please see lines 271-274).

Comment: *Whilst there is a fairly comprehensive review and integration of previous literature, I would suggest that the work of Fulweiler et al (2011 - DOI 10.1007/s12237-011-9432-4) and Rees et al (2016 -<http://dx.doi.org/10.1016/j.dsr2.2015.12.006>) should both be incorporated into discussions here. Fulweiler’s work showed that for estuarine/coastal conditions there was an inverse relationship between pH and nitrification, that is the opposite effect to that shown here. Whilst Rees showed that N₂O concentration decreased with increasing OA and saw no change in numbers of AOA both of which are different to here. The study of Rees was in open water and in very different conditions to this paper, but as one of very few papers on this subject probably offers some discussion points pertinent to the control of N₂O release from nitrification.*

Response: Thank you very much for this suggestion. In the revised manuscript, these recommended studies have all been incorporated into the discussions: *“Whilst nitrification rate was reported to increase along a decreasing natural gradient of pH in Narragansett Bay⁴³, it was likely due to a combination of biogeochemical conditions rather than the effect of acidification alone”* (Please see lines 116-118); *“However, Rees et al.⁴⁴ documented inhibition of N₂O production by ocean acidification in cold temperate and polar seawaters. Assuming that nitrification is the main N₂O production pathway in their study, the response of the N₂O production to acidification would be different in polar seas”* (Please see lines 171-174). In addition, Rees’s work showed that AOA assemblage composition lacked sensitivity to short-term ocean acidification, and this point has also been incorporated into the discussions: *“However, Rees et al.⁴⁴ reported that AOA assemblage composition was not sensitive to ocean acidification, possibly because the incubation period (less than one week) in their study was not long enough to cause significant turnover in the assemblage”* (Please see lines 387-390).

Specific comments:

Comment: L33 Whilst offering important insights the claim that this work will “help predict climate change” is taking things too far.

Response: As suggested, “help predict climate change” has been removed from the revised manuscript, and this sentence has been changed to “*This study highlights the molecular underpinnings of acidification effects on nitrification and associated greenhouse gas N₂O emission, and helps predict the response and evolution of estuarine and coastal ecosystems under climate change and human activities*” (Please see lines 33-37).

Comment: L121-122 eg Fulweiler et al.

Response: It has been incorporated in the revised manuscript, as suggested. Please see lines 123-124.

Comment: L129 disturbance on nitrification COULD cause profound consequences

Response: As suggested, “would” has been modified to “could” in the revised manuscript. Please see line 135.

Comment: L133 Beman predicted, Rees et al confirmed for open ocean conditions

Response: As suggested, these discussions have been provided in the revised manuscript. Please see lines 171-174.

Comment: L234-235 There was NOT enough mRNA with good integrity

Response: This point has been modified in the revised manuscript, as suggested. Please see lines 249-250.

Comment: L245 I'm not convinced that the data shown in Supplementary Fig 8 shows that steady state was reached after 25 days. There needs to be some informed discussion on this figure. I can't see why ammonium concentration in “Influent” was as high as 60µM, the ambient at Yz3 was ~10µM, but then where does it go, as is close to zero in effluent – surely not all is nitrified??

Response: In the revised manuscript, “steady state” has been modified to “*After approximately 25 days, the continuously operated simulators exhibited stable nitrification reactions (Supplementary Fig. 10) and dominance of nitrifying communities (accounting for 44.6% and 45.5% in the ambient controls and acidified treatments, respectively)*” (Please see lines 259-262). In addition, at the beginning of

the long-term incubation, NH_4^+ concentration of water samples in the manipulation systems was supplemented to a final concentration of approximately $10 \mu\text{M}$. After about half of the NH_4^+ was consumed (day 10), filter-sterilized site water with proper NH_4^+ concentrations (gradually increased from $20 \mu\text{M}$ to $60 \mu\text{M}$ with the increase of NH_4^+ consumption rate) were used as culture medium and supplied at a flow rate of 1 L day^{-1} to all reactors (Supplementary Fig. 10). Since the influent culture medium was continuously added to the manipulation systems drop by drop, the added NH_4^+ can be rapidly consumed and thus the NH_4^+ concentration in the effluent was close to zero. These explanations have been added to Supplementary Fig. 10, due to the word limit in the main text.

Comment: L218-224. *This is hard to follow and needs re-writing.*

Response: In the revised manuscript, it has been changed to “*It was previously speculated that increasing levels of $p\text{CO}_2$ may cause positive effect on the activity of chemoautotrophic nitrifiers⁵⁴. In contrast, under acidified conditions, elevated $p\text{CO}_2$ may further inhibit nitrification rate and promote the undesirable by-product N_2O emission. Therefore, the negative effects of aquatic acidification on microbial nitrogen transformations and their feedback to global climate change are probably more intensive than previously thought (Supplementary Fig. 8)*” (Please see lines 233-239).

Comment: L226 *Transcriptional response of nitrifiers to acidification during long-term incubation.*

Response: It has been modified as suggested. Please see line 241.

Comment: L278 *Supplementary Fig 15 referred to before 12, 13, 14.*

Response: The figures have been correctly ordered in the revised manuscript.

Comment: L564 *nutrient analysis likely to be colorimetrically rather than spectrophotometrically.*

Response: In the revised manuscript, “spectrophotometrically” has been replaced by “colorimetrically”. Please see line 624.

Comment: Fig 3 *How do “different lowercase letters above the columns indicate significant differences”?*

Response: Different lowercase letters above the columns (e.g., “a” vs. “b” or “a” vs. “c”) indicate that significant difference was detected between these two groups. In contrast, if two groups have the same lowercase letter (e.g., the groups of $800 \mu\text{atm/pH}$

8.1 and 400 $\mu\text{atm/pH}$ 7.8 in the right figure, both labeled with “b”), there is no significant difference between them. This method has been widely used to show statistical difference.

Comment: *Supp Fig 9 b – Typo Environment.*

Response: It has been modified in the revised manuscript.

Comment: *Supp Fig 14 & 16 – no reference to this figure found in text.*

Response: In the revised manuscript, they were all cited in the Main Text.

Comment: *Supp Table 9, 10, 11 – no reference to this table found.*

Response: They were all cited in the Main Text of the revised manuscript.

Reviewer #3 (Remarks to the Author):

Comment: *This paper addresses an important issue relating to climate change and environmental impacts in the coastal zone. Authors used a suite of technologies to investigate nitrification and emission of nitrous oxides under ocean acidification, overall the paper has potential in getting published in Nature Communications, but have the following concerns.*

Response: Thank you very much for your comments. This manuscript has been carefully modified according to your suggestions, and all the raised issues have been addressed as follows.

Major comments:

Comment: *1. Although the authors distinguished the effects of acidification from $p\text{CO}_2$ and pH only, acidification decreased nitrification rate and increased N_2O production. This main result has been shown in a previous study (<https://doi.org/10.1038/s41558-019-0605-7>).*

Response: In the study of Breider et al. (2019), they reported that when seawater pH in the western North Pacific was reduced, the N_2O production during nitrification increased significantly while nitrification rates remained stable or decreased. However, **(1)** “in their work, the acidification was manipulated by adding strong acid (HCl). Although the addition of HCl can elevate $p\text{CO}_2$ and reduce pH, it also alters alkalinity and results in different carbonate parameters compared with those expected in the future, i.e., dissolved inorganic carbon (DIC) increases under natural aquatic acidification rather than remains unchanged⁴⁶” (Please see lines 145-149). Therefore, further studies that can best mimic the ongoing aquatic acidification are still required.

(2) Compared with open ocean, the environmental conditions as well as the nitrifier communities are quite different in the complex estuarine and coastal waters. For example, the ammonium concentration could be several orders of magnitude higher in the estuarine and coastal regions due to intensive human activities, and *“Moreover, nitrifying communities in diverse aquatic habitats may respond differently to acidification, as the mechanisms for N₂O production differ among different nitrifying organisms^{32,38,40}. Therefore, the response of N₂O production during nitrification to aquatic acidification in estuarine and coastal waters needs to be evaluated”* (Please see lines 150-153). (3) Furthermore, this study explores the mechanisms of acidification affecting nitrification by decoupling the individual effects of elevated pCO₂ and reduced pH: *“This is the first attempt to decouple the individual effects of elevated pCO₂ and reduced pH on nitrification rate and associated N₂O emission in acidified aquatic environments, providing new insights into the underlying mechanism of aquatic acidification affecting nitrifiers”* (Please see lines 228-231). (4) More importantly, the transcriptional responses of nitrifying communities to aquatic acidification was revealed via metatranscriptomic analyses. This is a further step than Breider et al. (2019). Overall, this research provides insights into the underlying mechanisms of acidification affecting nitrification, and helps evaluate and predict the future ecological evolution of estuarine and coastal ecosystems. Related information has been further clarified in the revised manuscript.

Comment: 2. *I was confused by the sampling strategy. It seems that the bottom (sedimentary) water was taken? why not sample the surface water that is mainly affected by acidification. Also, it is widely accepted that nitrification functions well in the surface, and it is less likely in the bottom water.*

Response: Thank you very much for this comment. Unlike ocean acidification which is mainly driven by elevated atmospheric CO₂ concentrations, estuarine and coastal water is also significantly affected by land-derived inputs and complex biogeochemical and hydrological processes (Scanes et al., 2020). One of the greatest threats to estuarine and coastal ecosystems worldwide is the excess nitrogen loading and consequent eutrophication (Al-Haj & Fulweiler, 2020). The eutrophication-induced phytoplankton production can result in high respiration rate and strong CO₂ production in bottom waters (Laurent et al., 2017). Thus, near-bottom water generally suffers from more severe acidification in estuarine and coastal regions (Guo et al., 2021).

In addition, it was previously reported that nitrification is in general more active near the bottom of the shelf region (<60 m) (Shiozaki et al., 2019) or near the bottom of the euphotic zone of the open oceans (approximately 50~250 m) (Beman et al., 2011;

Rees et al., 2016; Breider et al., 2019). Similarly, in the Yangtze Estuary and adjacent coastal waters, higher nitrification rates were generally observed in the bottom water (Wang et al., 2018). Therefore, near-bottom water samples (water depth: 9~31 m) were collected in the present study. This point has been clarified in the revised manuscript: “*In this study, near-bottom waters, which typically exhibit more active nitrification and more severe acidification^{65,66}, were collected from these sites...*” (Please see lines 433-434).

Comment: 3. Why not consider Comammox process, not only on transcript level? I think this is important when evaluating nitrification process. Did authors conduct amplicon sequencing?

Response: Thank you very much for this comment. Actually, the newly discovered complete ammonia oxidizers (comammox) were considered in the present study. The abundance of comammox bacteria, as well as the canonical ammonia oxidizers: ammonia-oxidizing bacteria (AOB) and ammonia-oxidizing archaea (AOA) were quantified via qPCR. Results showed that the communities of the ammonia oxidizers changed significantly from the estuary mouth to its adjacent coastal area (Supplementary Fig. 3). AOB were the dominant ammonia oxidizers in the upper estuary waters whereas the adjacent coastal regions were dominated by AOA. However, the abundance of comammox bacteria was low in the estuarine and coastal waters, which were only detected at the upper estuarine sites (Yz1-Yz3) but not in the adjacent coastal waters (Yz4-Yz6) (Supplementary Fig. 3), suggesting that comammox bacteria may play an insignificant role in estuarine and coastal waters. In the present study, based on environmental water samples containing *in situ* nitrifying communities (including comammox bacteria), the responses of nitrification rate and associated N₂O production to acidification were investigated.

In addition, amplicon sequencing was conducted to explore the whole nitrifying communities, not just comammox bacteria, using the universal primers capable of detecting both bacteria and archaea within the same sequencing libraries. Furthermore, the transcriptional responses of the key nitrifying genes (*amo*, *hao*, *nxr*) of comammox bacteria to acidification were mined from the metatranscriptome data: “*In addition, based on metatranscriptome data, higher comammox gene transcripts were detected in the acidified treatments (0.78-11.59% of the total obtained gene transcripts involved in the stepwise oxidation of NH₃ to NO₃⁻) compared with the ambient control (only 0.03-0.89%, P<0.05) (Fig. 6b and Supplementary Fig. 17)*” (Please see lines 398-402).

Minor comments:

Comment: 1. what does "CO₂-fertilization" mean? This is mainly used to describe this effect in terrestrial ecosystem. It should be avoided using any ambiguous words in abstract;

Response: The explanation of "CO₂-fertilization" has been provided in the revised manuscript: "Higher pCO₂ condition is expected to benefit nitrification, as an increased carbon source may promote the growth of chemoautotrophic nitrifiers (CO₂-fertilization)²⁴⁻²⁷" (Please see lines 67-69). In many previous studies, "CO₂-fertilization" has been used to describe the effect of increased CO₂ on chemoautotrophs in aquatic habitats (e.g., Hutchins et al., 2009, Beman et al., 2011; Rees et al., 2016; Breider et al., 2019). To avoid ambiguity in the abstract, an explanation has also been provided as suggested: "beneficial effect of elevated CO₂ on activity of nitrifiers ("CO₂-fertilization" effect)" (Please see lines 30-31).

Comment: 2. what does "efficient changes" in gene expression refer to in the abstract? This sentence here does not mean much sense, it does not convey any practical info;

Response: Thank you very much for this comment. As suggested, "make efficient changes in gene expressions to cope with acidification stress" has been changed to "nitrifiers could significantly up-regulate gene expressions associated with intracellular pH homeostasis to cope with acidification stress" (Please see lines 32-33).

Comment: 3. Misleading statement. "However, how microbially-mediated biogeochemical processes may be altered by acidification in estuarine and coastal ecosystems remains largely unknown, to a great extent limiting our understanding of the responses of these ecosystems to global environmental changes". Many studies have explored the microbially-mediated biogeochemical processes in estuarine and coastal ecosystems. E.g., DOI10.1038/s41561-020-0584-3; DOI10.1021/acs.est.2c00692; DOI10.1002/2014GL060849; DOI10.1016/j.scitotenv.2020.142689; DOI10.1038/s41467-017-00417-7

Response: As suggested, this statement has been modified, and the recommended researches have been included in the revised manuscript: "Acidification in estuarine and coastal waters can thus be greatly intensified by episodic intrusion of high-CO₂ upwelled water^{11,13,14}, which may detrimentally affect biological processes and functioning of estuarine and coastal ecosystems¹⁵⁻²¹" (Please see lines 56-59).

Comment: 4. Line 95, delete "new";

Response: It has been removed from the revised manuscript, as suggested.

Comment: 5. Line 377 Indeed, the comammox *amoA* gene abundance was higher in the acidified treatments, how did the authors confirm that this *amoA* gene is affiliated to comammox, not for AOA or AOB?

Response: Copy numbers of comammox *amoA* gene were quantified by qPCR using the specific primer set A378F/C616R designed for comammox bacteria, and the specificity of this primer set was confirmed using clone libraries (Xia et al., 2018). In addition, taxonomic affiliations to the *amoA* gene transcripts were assigned by binning to the best hit in the NR database (BLASTP, $e\text{-value} \leq 10^{-5}$) and were further confirmed by the phylogenetic analyses. Based on the constructed maximum-likelihood phylogenetic tree of *amoA* gene, the gene sequences affiliated to comammox bacteria can be clearly identified. In the revised manuscript, the phylogenetic tree of *amoA* gene has been provided as Supplementary Fig. 17.

Comment: Overall, I think work is interesting and authors should improve the MS to distill the major significance better, as opposed to previously reported results, as indicated earlier.

Response: Thank you very much for your comments. As shown above, the manuscript has been greatly improved to better distill the major significance according to all these comments and suggestions.

References:

- Beman, J. M. et al. Global declines in oceanic nitrification rates as a consequence of ocean acidification. *Proc. Natl Acad. Sci. U. S. A.* **108**, 208-213 (2011).
- Breider, F. et al. Response of N₂O production rate to ocean acidification in the western North Pacific. *Nat. Clim. Change* **9**, 954-958 (2019).
- Dalsgaard, T. et al. Oxygen at nanomolar levels reversibly suppresses process rates and gene expression in anammox and denitrification in the oxygen minimum zone off northern Chile. *mBio* **5**, e01966-14 (2014).
- Doney, S. C. The growing human footprint on coastal and open-ocean biogeochemistry. *Science* **328**, 1512-1516 (2010).
- Guo, X. H. et al. Seasonal variability and future projection of ocean acidification on the east China sea shelf off the Changjiang Estuary. *Front. Mar. Sci.* **8**, 770034 (2021).
- Hutchins, D. A. & Capone, D. G. The marine nitrogen cycle: new developments and global change. *Nat. Rev. Microbiol.* **20**, 401-414 (2022).
- Laurent, A. et al. Eutrophication-induced acidification of coastal waters in the northern Gulf of Mexico: Insights into origin and processes from a coupled physical-biogeochemical model. *Geophys. Res. Lett.* **44**, 946-956 (2017).

- Rathnayake, R. M. L. D. et al. Source identification of nitrous oxide on autotrophic partial nitrification in a granular sludge reactor. *Water Res.* **47**, 7078-7086 (2013).
- Rees, A. P., Brown, I. J., Jayakumar, A. & Ward, B. B. The inhibition of N₂O production by ocean acidification in cold temperate and polar waters. *Deep-sea Res. Pt. II* **127**, 93-101 (2016).
- Scanes, E., Scanes, P. R. & Ross, P. M. Climate change rapidly warms and acidifies Australian estuaries. *Nat. Commun.* **11**, 1803 (2020).
- Shiozaki T. et al. Factors regulating nitrification in the Arctic Ocean: potential impact of sea ice reduction and ocean acidification. *Glob. Biogeochem. Cycles* **33**, 1085-1099 (2019).
- Sutka, R. L. et al. Distinguishing nitrous oxide production from nitrification and denitrification on the basis of isotopomer abundances. *Appl. Environ. Microbiol.* **72**, 638-644 (2006).
- Wang, W. T. et al. Rates of nitrification and nitrate assimilation in the Changjiang River estuary and adjacent waters based on the nitrogen isotope dilution method. *Cont. Shelf Res.* **163**, 35-43 (2018).
- Xia, F. et al. Ubiquity and diversity of complete ammonia oxidizers (comammox). *Appl. Environ. Microbiol.* **84**, e01390-18 (2018).